

# Improvements of the land surface configuration to better simulate seasonal snow cover in the European Alps with the CNRM-AROME (cycle 46) convection-permitting regional climate model

Diego Monteiro[1], Cécile Caillaud[2], Matthieu Lafaysse[1], Adrien Napoly[2], Mathieu Fructus[1], Antoinette Alias[2], and Samuel Morin[2]

[1]Univ. Grenoble Alpes, Université de Toulouse, Météo-France, CNRS, CNRM, Centre d'Etudes de la Neige, 38000 Grenoble, France
[2]CNRM, Météo-France, CNRS, Université de Toulouse, Toulouse, France

**Correspondence:** Diego Monteiro (diego.monteiro@meteo.fr)

**Abstract.**

Snow cover modeling remains a major challenge in climate and numerical weather prediction (NWP) models, even in recent versions of high-resolution coupled surface-atmosphere (i.e. at km-scale) regional models. Evaluation of recent climate simulations, carried out as part of WCRP-CORDEX Flagship Pilot Study on Convection with the CNRM-AROME convection permitting regional climate model at 2.5 km horizontal resolution, has highlighted significant snow cover biases, severely limiting its potential in mountain regions. These biases, which are also found for AROME NWP model results, have multiple causes, involving atmospheric processes and their influence on input data to the land surface models, in addition to deficiencies of the land surface model itself. Here we present improved configurations of the SURFEX-ISBA land surface model used in CNRM-AROME. We thoroughly evaluated these configurations on their ability to represent seasonal snow cover across the European Alps. Our evaluation was based on coupled simulations spanning the winters of 2018-2019 and 2019-2020, which were compared against remote sensing data and in situ observations. Specifically, the study tests the influence of various changes to the land surface configuration, such as using a multi-layer soil and snow scheme, multiple patches for land surface grid cells, new physiographic databases, and parameter adjustments. Our findings indicate that using more physically detailed individual components in the surface model using only one patch did not improve the representation of snow cover due to limitations in the approach used to account for partial snow cover within a grid cell. To address these limitations, we evaluated further configurations using three patches and improved representations of the interactions between fractional snow cover and vegetation. At the end, we introduce a land surface configuration that substantially improved the representation of seasonal snow cover in the European Alps. This holds promising potential for the use of such model configurations in climate simulations and numerical weather prediction, including AROME and other high-resolution climate models.



## 1 Introduction

Accurate modeling of land surface-atmosphere interactions in mountainous regions is crucial for numerical weather prediction (NWP) and climate projections. Applications range from short-term forecasts for weather-dependent human activities (risk prediction, hydropower production, tourism and traffic management) to long-term studies of the impacts of climate change on the various components of mountain range. On these regions, an appropriate representation of the seasonal snow cover is crucial, as its presence strongly affects the evolution of the surface and near-surface conditions, by the modification of the albedo, the roughness of the terrain, and its insulating properties on the underlying soil.

Snow models used in NWP and climate studies are widely evaluated and tested in stand-alone configurations at the point scale (Decharme et al., 2016; Menard et al., 2020). However, when used in coupled surface-atmosphere simulations, these models are likely to produce significantly different results due to errors arising from atmospheric modeling (Raleigh et al., 2015; Lapo et al., 2015), the use of sub-grid parameterizations to account for surface heterogeneity inside a grid cell of a discretized domain, and their own deficiencies. Assessing the representation of snow in coupled configurations is a necessary complementary approach to standalone model run. This is however a difficult task especially over complex terrain.

Modeling atmospheric and surface fluxes as well as snow cover in mountain regions is challenging in many aspects. The complex topography induces a number of phenomena on a wide range of spatio-temporal scales (e.g. Föhn effect, convection phenomena, preferential deposition of snowfall, temperature inversions and snow redistribution processes) which have a major impact on surface weather conditions. In addition, the strong heterogeneities of the surface characteristics (elevation, surface type and aspect) generate high variability of near-surface conditions at sub-kilometer scales, affecting all surface components, including the snowpack. The high variability of atmospheric and surface conditions in mountains at a fine scale suggests that the use of high-resolution models would minimize modeling uncertainties, by limiting the use of sub-grid parameterizations.

Most regional coupled atmosphere-surface climate models (RCMs) exhibit deviations with respect to observational references, which can be particularly substantial in mountainous areas. In the European Alps, numerous studies have evaluated the EURO-CORDEX regional climate simulations ensembles at a horizontal resolution from 12 to 50 km, and identified strong biases in near-surface precipitation and temperature indicators (Kotlarski et al., 2014; Smiatek et al., 2016; Vorkauf et al., 2021), showing generally excessive precipitation and too low temperature values. In general, snow cover (depth, mass, duration) is overestimated (Terzago et al., 2017; Matiu et al., 2020). One potential approach to mitigate these issues is to develop and apply kilometer-scale modelling frameworks, such as the WCRP-CORDEX Flagship Pilot Study (FPS) on Convection (Coppola et al., 2020; Ban et al., 2021; Pichelli et al., 2021), which have great potential for mountain regions.

At Météo-France, the limited-area non-hydrostatic model AROME (Applications de la Recherche à l'Opérationnel à Méso-Echelle, Seity et al., 2011; Brousseau et al., 2016) has been used operationally for NWP since 2008, initially at 2.5 km horizontal resolution, 1.3 km since 2015, and used for climate studies, referred to as CNRM-AROME, since 2014 (Déqué et al., 2016; Fumière et al., 2019; Caillaud et al., 2021), at 2.5 km horizontal resolution. Simulations results of these models exhibit a number of issues that limit their use and relevance in mountain regions. Indeed, in a recent study comparing 30 years of past climate simulations carried out with CNRM-AROME with the S2M reanalysis (Vernay et al., 2022) over the French Alps, we





highlighted a negative temperature difference on the order of 2 to 3°C, maximum in winter at high elevations, and an excess
amount of precipitation, particularly at high elevations (Monteiro et al., 2022).

In these climate simulations, we were also able to identify substantial snow cover biases, such as an excessive snow accumulation at intermediate and high elevations, with an overestimated snow cover extent and duration (Monteiro and Morin, 2023), and unrealistic snow accumulation on some grid cells, reaching several hundred meters after 30 years of simulation. A near surface temperature bias has also been identified and analyzed in the NWP version of AROME (Vionnet et al., 2016;
Arnould et al., 2021; Gouttevin et al., 2023). These dismiss issues related to the horizontal resolution of the model, but rather pointed towards multiple other factors, namely the underestimation of the cloud cover, also identified by Lucas-Picher et al. (2023) in CNRM-AROME climate simulations, the underestimation of turbulent mixing under stable conditions, and a strongly underestimated sub-surface soil temperatures used to diagnose the near surface air temperature.

The origin of the widespread overestimation of snow amount and duration in AROME model results is certainly multiple.
Monteiro et al. (2022) identified several factors.

- Biased atmospheric forcings, such as an overestimation of snowfall and an underestimation of melting due to excessively cold near-surface temperatures and errors in downard radiation fluxes.

- The use of an overly simplified surface configuration (1-layer snow model and force-restore soil scheme).

- The lack of glacier dynamics and snow redistribution processes, leading to the creation of "snow towers" on some high
elevation grid points (Freudiger et al., 2017).

The land surface configuration used in the current version of the CNRM-AROME model (also in used in the current version of AROME used for NWP applications) does not provide an adequate representation of snow cover dynamics over the French and European Alps, with potential effects on other surface variables of interest such as the near surface air temperature.

In this study, we aim to investigate the representation of snow cover for a set of surface model configurations already
implemented within the land surface model SURFEX-ISBA (Noilhan and Mahfouf, 1996; Masson et al., 2013) but not yet evaluated in a coupled surface-atmosphere context at high resolution, such as CNRM-AROME, in mountainous regions.

In this context, we document the advantages and limitations of using different levels of complexity in the representation of the snowpack and the soil: from a single-layer parameterization for snow (Douville et al., 1995) and a "force-restore" scheme for soil (Noilhan and Mahfouf, 1996), to explicit multi-layer modules for both snow (Boone and Etchevers, 2001) and soil
(Boone et al., 2000; Decharme et al., 2011). In addition to improving the individual components of the model (soil and snow schemes), we test the use of multiple patches (i.e. a "tiling approach") to divide the energy balance by surface types, even for kilometer scale modeling systems, and address limitations of sub-grid parametrisations such as the partial snow cover fraction approach when only one soil column is used for both covered and uncovered snow parts. As these approaches are common to many LSMs used in coupled systems (e.g. HTESSEL (Balsamo et al., 2009), NOAH-MP (Niu et al., 2011), CLM5
(Lawrence et al., 2019), JULES (Best et al., 2011)), our study may provide information on the necessary content of surface configurations to correctly represent snow cover in mountainous regions in a high-resolution coupled surface-atmosphere



context. The identify shortcomings may also explained some of the snow cover issues raised in coupled systems at coarser resolutions using SURFEX-ISBA LSM, such as CNRM-ALADIN (Termonia et al., 2018) in the Alps (Monteiro and Morin, 2023), as well as CNRM-CM6 in high-latitude boreal forests (Decharme et al., 2019).

The results of these experiments are analyzed and compared to different sets of observational data, enabling us to assess the impact of the modifications in complementary ways:

- Comparisons of snow depth values on a large set of in-situ measurements collected and presented in Matiu et al. (2021a), enabling quantitative analysis on a broad spatial scale.

- Comparisons with MODIS snow durations, providing near-exhaustive spatial coverage of time-aggregated information
on snowpack conditions.

In the end, we introduce a SURFEX-ISBA configuration that is relevant for coupled surface-atmosphere modeling and allows for a significant improvement in the representation of mountain snow cover.

## 2   Materials and methods

### 2.1   CNRM-AROME model

In this study, simulations are carried out using the CNRM-AROME climate model, which is the convection-permitting regional climate model (CP-RCM) used at CNRM, which includes the surface model SURFEX (SURFace EXternalisée) (Masson et al., 2013) coupled to the AROME atmospheric model. CNRM-AROME is directly based on the non-hydrostatic limited-area model AROME used for NWP at Météo-France since 2008 (Seity et al., 2011; Brousseau et al., 2016). An alternative version of the AROME model referred to as HARMONIE-AROME (Bengtsson et al., 2017) is used in NWP applications by several European
meteorological services, also used for climate studies by the HARMONIE-CLIMATE community (Belušić et al., 2020; Lind et al., 2020).

In this study, the CNRM-AROME model is based on NWP AROME cycle 46t1 in operational use at Météo-France since 2022, operated for climate simulations at a horizontal resolution of 2.5 km with 60 vertical levels. The timestep of the model is 60 s. This version has much in common with cycle 41t1, used for the CNRM-AROME climate simulations carried out as part
of FPS convection of CORDEX (Coppola et al., 2020; Pichelli et al., 2021). Detailed information about its atmospheric and surface configuration can be found in Termonia et al. (2018); Caillaud et al. (2021). The main differences between the cycle 41t1 and 46t1 relevant to our study is the use of a more recent version of SURFEX (version 8.0).

### 2.2   SURFEX : the surface platform

For this study, the surface modeling is ensured by the surface platform SURFEX v8.0 (Masson et al., 2013). Within SURFEX,
the estimation of energy and mass fluxes of each grid cell is carried out by specific modules depending on the type of surface environments, called tiles. Four distinct such environments are accounted for in SURFEX :



  – Tile NATURE : "natural" continental surfaces (i.e. including bare soil, rocky ground, permanent snow, glaciers, natural and cultivated vegetation), using the ISBA Land Surface Model (LSM) (Noilhan and Planton, 1989; Noilhan and Mahfouf, 1996),

– Tile TOWN : urban environments, using the TEB module (Masson, 2000),

  – Tile LAKE : continental water bodies such as lakes and rivers using the Charnock formulation (Charnock, 1955),

  – Tile SEA : seas and ocean, using the version 6 of ECUME (Belamari and Pirani, 2007),

The NATURE land surfaces modeling is carried out by the LSM ISBA, representing the evolution of soil and vegetation biophysical variables, including the snowpack, either parameterized or explicitly represented.

### 2.2.1 The ISBA LSM : main principles and identified weaknesses/flaws for snow representation

Three main different land surface configurations are described and analysed in the study. Despite their differences, the calculation of the surface energy balance and the parameterization of the snow fraction are identical, and play a major role in the seasonal evolution of the snowpack.

**Surface energy balance**

The surface energy balance is computed for a surface layer with a fixed depth of 0.01 m, which is a composite representation of the soil-vegetation system (soil-vegetation-snow in the case of the approach using the D95 single-layer snow parameterization (Douville et al., 1995)).

A single surface temperature $Ts$ is calculated for each grid cell, whose evolution depends on the surface heat flux into the composite layer $G$, the heat flux between the surface and the soil $F_{surface-soil}$, for which the formulation depends on the soil 135 scheme used, and the heat flux between the surface and the snowpack $F_{surface-snow}$, in the case of the use of an explicit snow model.

$$\frac{dTs}{dt} = Ct \times G - F_{surface-soil} - F_{surface-snow} \tag{1}$$

with $G$ ($W\,m^{-2}$) an energy flux resulting from the evolution of the radiation balance $Rn$ and the sensible $H$ and latent $L$ heat fluxes, weighted by $Ct$, a composite coefficient accounting for the heat capacity of the surface layer, whose formulation 140 depends on the soil and snow scheme used.

$$G = Rn - H - LE \tag{2}$$

The radiation balance is the cumulated difference ($W\,m^{-2}$) between the incoming shortwave $SWd$ and longwave $LWd$, and the outgoing shortwave $SWu$ and longwave $LWu$.





$$Rn = SWd + LWd - SWu - LWu \qquad SWu = SWd \times \alpha_s \qquad LWu = \varepsilon_s \sigma Ts^4 \tag{3}$$

with $\alpha_s$ and $\varepsilon_s$ respectively the surface albedo and emissivity, and $\sigma$ the Stefan-Boltzman constant.

The turbulent fluxes are computed by means of the bulk aerodynamic formulae defined by Louis (1979), and modified by Mascart et al. (1995) to account for different roughness length values for heat and momentum.

**The patch approach**

In order to take into account the heterogeneity of the land surface within the NATURE tile of each model grid cell, ISBA offers

the possibility to split the calculation of energy balances by surface types. A total of 19 surface types (called patches) are available, dividing natural surfaces into soil and vegetation categories with distinct physical characteristics. The nomenclature and categorization of the 19 patches are taken from the ECOCLIMAP physiographic database (Faroux et al., 2013) and correspond to the Plant Fonctional Types (PFTs) of ECOCLIMAP. In this study, we use ECOCLIMAP version I (Masson et al., 2003).

The number of patches is set by the user, with a number ranging from 1 to 19. When less than 19 patches are used, the physical

characteristics of multiple land surfaces are aggregated, by grouping them by categories, and weighted by their respective fractions within the cell while following the aggregation laws defined by Noilhan et al. (1995) and Noilhan et al. (1997) (e.g. logarithmic for the roughness length, linear for the albedo, the Leaf Area Index (LAI) and the vegetation fraction, inverse for the stomatal resistance).

For a given grid cell, the atmospheric fluxes received are thus identical for all tiles and patches, but a specific energy and

mass balance is calculated for each of the patches. There is not any energy and mass exchanges between the soil-snow columns of the different patches. The fluxes for each of the patches are then aggregated by weighting by the relative fraction of each type of surfaces within the grid cell, enabling the estimation of average fluxes for all the natural surface types in the grid cell, which are provided to the atmospheric model or used as diagnostics for each grid point.

**Parameterization of the snow cover fraction**

The presence of snow on the ground has a major impact on the surface mass and energy balance in several ways. As the snow cover extends, the albedo of the surface increases, its roughness decreases, and the snowpack insulates the underlying ground from heat and mass exchanges with the atmosphere. The way in which the fraction of the grid cell covered by snow is calculated and influence the computation of the energy balance is therefore critical and is represented in widely different ways in different land surface models (Essery et al., 2013; Menard et al., 2020; Lalande et al., 2023).

In ISBA, for each patch the snow fraction is calculated differently depending on whether vegetation is present or not, and the fraction is used for energy balance calculations.

In the absence of vegetation, even a small amount of snow covers the entire surface. This is represented in ISBA by the fact that the snow cover fraction in non-vegetated areas ($P_{sng}$) reaches 1 as soon as the snow water equivalent $Ws$ exceeds a threshold value set to $Ws_{crit} = 10 \, \mathrm{kg \, m^{-2}}$ (Equation (4)).





$$P_{sng} = min(1, \frac{Ws}{Ws_{crit}}) \tag{4}$$

In the presence of vegetation, the calculation of the snow cover fraction ($P_{snv}$) is done based on the snow depth (also referred to as the height of the snow) value $Hs$ and takes into account the height of the vegetation through the roughness length $z_0$. $W_{sn}$ is a scaling factor, modulating the weight of vegetation height in the calculation of the snow cover fraction (Equation (5)).

$$P_{snv} = \frac{Hs}{(Hs + W_{sn}z_0)} \tag{5}$$

The total snow cover fraction $P_{sn}$ is the sum of the snow fractions for each patch weighted by their respective fraction (Equation (6)).

$$P_{sn} = (1 - Veg) \times P_{sng} + Veg \times P_{snv} \tag{6}$$

Snow-related prognostic variables are defined for each patch. Integrated diagnostics for each grid cell (in fact, the NATURE tile of each grid cell) are computed as the weighted average using the patch fractions.



## 2.3 Land surface configurations


The objective of the study is to describe and evaluate new land surface configurations, in order to improve the representation of seasonal snow cover in the European Alps and address some of the issues identified in Monteiro et al. (2022). Consequently, the atmospheric configurations and initialization of all experiences are similar, and we explore the impacts of changes in surface configuration mostly on the simulated snowpack. Note also that, part of the content of the configurations tested here were already used in the latest version of the General Circulation Model (GCM) CNRM-CM6 (Decharme et al., 2019) and the RCM CNRM-ALADIN63 (Nabat et al., 2020) but had not been used in coupled model simulations using AROME.


For all configurations tested in this study, including the configuration referred to as the default one, we activate the option, described in Decharme et al. (2016), that limits snow accumulation above a certain threshold (see Decharme et al. (2016)). Its value is set to the default value at 33.3m. This option is activated in all experiments and avoids the formation of "snow towers" problem identified in Monteiro et al. (2022).


Figure 1 illustrates the main characteristics of the three main surface configurations used in this study. The configurations are described in detail in sections 2.3.1, 2.3.2 and 2.3.3 and Table 1 summarize their main model components.



| Model features | Configurations | | |
|---|---|---|---|
| | **D95-3L** | **ES-DIF** | **ES-DIF-OPT** |
| **Soil** | Force restore: two layers (3L) - (Boone et al., 1999) | Explicit multilayer scheme (DIF) - (Boone et al., 2000; Decharme et al., 2011) | Explicit multilayer scheme (DIF) - (Boone et al., 2000; Decharme et al., 2011) |
| **Snow processes** | Single layer bulk snow model - (Douville et al., 1995) | Intermediate complexity: ISBA-ES - (Boone and Etchevers, 2001) | Intermediate complexity: ISBA-ES - (Boone and Etchevers, 2001) |
| **Patch number (see section 2.2.1)** | 1 | 1 | 3 |
| **Physiographic input dataset** | HWSD for soil texture | HWSD for soil texture | SoilGrids v2.0 for soil texture and soil organic carbon |
| **Snow fraction parameter WSN (see eq. 5 section 2.2.1)** | 5 | 5 | 1 |
| **Others** | | | + modification of the thermal conductivity of the soil-snow interface + activation of the parameterization of soil organic (see section 2.3.3) |
| **Computational time relative to the D95-3L configuration** | **1** | **1.15** | **1.25** |

**Table 1.** Main model components for each configuration







**Figure 1.** Schematic illustration of the main physical processes, flux exchanges and prognostic variables for the three main configurations documented and tested in this study. D95-3L configuration (orange framed), ES-DIF (blue framed) and the ES-DIF-OPT (green framed) with the different modifications displayed and schematically illustrated. $H$ and $LE$ : sensible and latent heat fluxes, respectively. $ROS$ : Rain on snow. $F_x$ : fluxes from the component $x$. $W_{xn}$ and $WI_{xn}$ : respectively the liquid and ice content of the $n_{th}$ layer of the component $x$, $s$ for surface, $g$ for ground, $sn$ for snow. $W_{sn}$ : Snow water equivalent. $\alpha_{sn}$ : snow albedo. $\rho_{sn}$ : snow density. $H_{sn}$ : snow enthalpy. $T_{sn}$ : snow temperature. $Wl_{sn}$, liquid content of the snow.



### 2.3.1   D95-3L : One layer snow parameterization, force-restore approach for the soil

This surface configuration is the default one as it is currently in use for NWP version of AROME and CNRM-AROME for
climate studies (Caillaud et al., 2021; Lucas-Picher et al., 2023; Monteiro et al., 2022). It is described schematically on Figure 1.
The evolution of biophysical soil variables is ensured by the "3-L" soil model, using a force-restore approach. Heat exchanges
in the ground (temperature evolution) are represented using two layers (Noilhan and Mahfouf, 1996), and three layers are used
for the evolution of hydrological variables (Boone et al., 1999). In this configuration, the snowpack is parameterized as a single
layer with homogeneous physical properties, referred to as the D95 parameterization (Douville et al., 1995), mixed with the
soil-vegetation composite surface layer. Consequently, no specific energy balance is solved for the snowpack, which is taken
into account by modifying the properties of the composite surface layer. The main equations governing the evolution of the
surface components are given below:

$$\frac{dTs}{dt} = Ct * G - F_{surface-soil} \tag{7}$$

$$\frac{1}{Ct} = \frac{(1-Veg)(1-P_{sng})}{Cg} + \frac{Veg(1-P_{snv})}{Cv} + \frac{P_{sn}}{Cs} \tag{8}$$

$$F_{surface-soil} = \frac{2\pi}{\tau}(Ts - Tg2) \tag{9}$$

with $Tg2$, the temperature of the deep soil layer (which evolves through a relaxation term towards $Ts$), $Cg$, $Cv$ and $Cs$
respectively the heat capacity of the ground, vegetation and snow.

Three prognostic variables characterize the snowpack:

- The density, an exponentially decreasing function, forced to $100 \, \mathrm{kg \, m^{-3}}$ for fresh snow, limited to $300 \, \mathrm{kg \, m^{-3}}$ for aged
  snow. The density of the entire snowpack is updated during snowfall by a weighted average of the layer previously
  present and that of the snow newly fallen to the ground.

- The albedo, whose evolution can follow two functions, forced to 0.85 for fresh snow, limited to 0.5 for old snow, linearly
  decreasing in the absence of melting and exponentially decreasing in the presence of melting (i.e. to account for wet
  metamorphism).

- The snow water equivalent (total mass) of the snowpack results from a mass balance calculation depending on snowfall,
  snow sublimation/evaporation and melting.

In this configuration, the snow layer has no prognostic temperature of its own, but is included in the composite soil-
vegetation-snow surface layer, from which the melting temperature, i.e. the temperature value used to compute the melt inten-
sity, is derived. In the presence of vegetation, the snow melt intensity is calculated based on a hybrid diagnostic temperature, a



weighted average between the surface soil temperature and the deep soil layer temperature, with a value closer to the deep soil
layer temperature as the proportion of vegetation increases (see equation 10).

$$T_{melt} = (1 - Veg)Ts + VegTp \tag{10}$$

with $T_{melt}$, the melting temperature, $Ts$, the instantaneous surface temperature, $Tp$, the daily mean surface temperature, and
$Veg$, the fraction of vegetation within the grid cell.

This approach was developed to prevent unrealistic snowpack melting. Indeed, using the instantaneous value of the surface
temperature representative of the soil-vegetation-snow system tend to be too high during daytime (i.e. due to the mixed albedo
between snow and vegetation), leading to spurious snowmelt computations (Douville et al., 1995). As shown later, this approach
has strong consequences for the modelling of snow conditions in forested environments.

### 2.3.2  ES-DIF : Multi-layer snow scheme, multi-layer soil scheme

This approach uses intermediate complexity schemes for soil and snow in a multi-layer manner, allowing the resolution of
specific energy balances for the soil-vegetation system and for snow, as well as a more detailed representation of the pro-
cesses within them. These are the schemes currently used in recent versions of the CNRM-CM6 global model (Voldoire et al.,
2019; Decharme et al., 2019), the CNRM-ALADIN regional model (Nabat et al., 2020), and in the most recent version of
the HARMONIE-Climate AROME regional climate model (Belušić et al., 2020; Lind et al., 2020). However, note that only
one patch is used herein for the NATURE tile, which is not the way the configuration is implemented for the coupled sys-
tems CNRM-CM6 and CNRM-ALADIN using 12 patches, and HARMONIE-Climate using 2 patches. This configuration is
illustrated in Figure 1.

Heat and mass exchanges within the soil are computed using the ISBA-Diffusion scheme (ISBA-DIF, Boone et al. (2000)
and Decharme et al. (2011)), with 14 layers from the surface to 12 m, representing explicitly heat exchanges within the different
soil layers through the resolution of a 1D Fourier law. In this scheme, a single surface temperature $Ts$ is calculated for the soil-
vegetation system, whose evolution depends on the surface energy balance ($G$, eq. 2), the surface-soil heat flux $F_{surface-soil}$
with the second soil layer on the non-snow-covered part, and the surface-snow heat flux at the surface-snow interface on the
snow-covered part.

The main equations are provided below:

$$\frac{dTs}{dt} = Ct \times G - F_{surface-soil} - F_{surface-snow} \tag{11}$$

$$\frac{1}{Ct} = \frac{(1 - Veg)(1 - P_{sng})}{Cg} + \frac{Veg(1 - P_{snv})}{Cv} \tag{12}$$

$$F_{surface-soil} = Cg\frac{\bar{\lambda}_1}{\Delta\bar{z}_1}(Ts - Tg2) \tag{13}$$





with $\bar{\lambda}_1$ $(W\,m^{-1}\,K^{-1})$the inverse-weighted arithmetic mean of the soil thermal conductivity at the interface between the surface layer and the underlying soil layer, $\Delta \bar{z}_1$ the thickness $(m)$ between the two consecutive layer mid-points. $F_{surface-snow}$

is a heat conduction term between the lowermost snow layer and the soil surface layer.

The snowpack evolution is carried out by the Explicit Snow scheme (ISBA-ES) (Boone and Etchevers, 2001; Decharme et al., 2016), using up to 12 snow layers, for which a specific energy balance is solved, unlike D95 whose energy balance is common to the composite soil-vegetation surface layer.

Three prognostic variables are used to describe the state of each layer at each time step:

– Heat content (i.e. temperature and water/ice content), which defines the energy required to melt the layer, and thus combines the information of snow temperature and liquid water content at melting point.

– Density, which evolves under the effect of parameterized compaction and metamorphism (Brun et al., 1989), wind-induced densification of near-surface snow layers and fresh snowfall (whose density is a function of air temperature and wind at the time of fall).

– The thickness of each layer, ranging from a few mm to several tens of cm, defined to be finest close to the ground/snow and atmosphere/snow interfaces (see Decharme et al. (2016) for more details).

One additional prognostic variables for the surface layer is the albedo. As stated by Boone and Etchevers (2001), snow albedo follows a linear decrease rate for dry snow (Baker et al., 1990), and an exponential decrease rate to model the wet metamorphism (Verseghy, 1991).

The mass balance of the snowpack is expressed as the sum of precipitation on snow (solid and liquid since each layer can have a liquid water content), evaporation and sublimation, as well as a term describing the flow of water out of the snowpack at its base.

ISBA-ES includes a number of parameterizations that reproduce the effects of physical processes affecting the evolution of the snowpack :

– Compaction, metamorphism and wind-induced densification (Brun et al., 1989).

– Transmission of incident solar flux through the layers (Brun et al., 1992).

– Water percolation between layers.

– Refreezing and melting of water contained in layers.

The temperature of all snow layers is computed simultaneous following this system of equations :

$$Cs_i D_i \frac{dTsn_i}{dt} = Gs_{i-1} - Gs_i + Rs_{i-1} - Rs_i - Ss_i \qquad (14)$$

with, for each layer $i$, $Ss_i$, representing the heat sink/source linked to water phase changes, $Rs_i$, the incident solar radiation transmitted (decreasing exponentially with distance to the snow surface), $Gs_i$, the layer energy balance, $Gs_{i-1}$, the energy





balance of the layer above. For the layers below the surface, the $Gsi$ term corresponds to thermal diffusion in snow, while for the uppermost layer the energy balance includes the following terms :

$$Gs_0 = Rns - H - LE - CwP_{sn}(P - Ps)(Tf - Tr) \qquad (15)$$

with $Rns$ the snow surface radiation balance, $H(Tsn_1)$ and $LE(Tsn_1)$ the turbulent fluxes above snow (calculated according to Louis (1979) formulae), and a latent heat source term related to the fall of liquid precipitation in the snowpack, with $Cw$, the heat capacity of water, $P$ and $Ps$ the total and solid precipitation respectively, $Tr$, the temperature of the rain and $Tf$ the fusion temperature. Any excess heating of snow temperature above the freezing point is converted in energy available for melting. Then, the liquid water percolation follows a bucket scheme based on a liquid water retention capacity, and accounting for possible refreezing in colder layers.

### 2.3.3 ES-DIF-OPT : Multi-layer snow scheme, multi-layer soil scheme, including optimal modifications

The ES-DIF-OPT configuration, stands for optimized ES-DIF configuration. It starts from the second configuration (ES-DIF) and adds a series of modifications (see figure 1) concerning the use of multiple patches, changes in some parameterizations, input physiographic databases and calculation of heat and mass exchanges.

**3-PATCHS**

The "3-PATCHS" modification consists in activating three patches for energy and mass balance calculations, in contrast to the D95-3L and ES-DIF configurations which only use one patch (see section 2.2.1 and illustration on figure 1c for further details). In the most recent version of the HARMONIE-Climate AROME model (2.5 km horizontal resolution) (Belušić et al., 2020), two patches have been activated, in order to distinguish between forest and open-land areas, while 12 patches are used in the latest version of CNRM-CM6 (150 km horizontal resolution) (Decharme et al., 2019) and CNRM-ALADIN (12.5 km horizontal resolution) (Nabat et al., 2020). When three patches are used, the categories are grouped into "uncovered surface" (e.g. permanent snow, rock, bare soil), "low vegetation" (shrubs, grass, crops) and "high vegetation" (various type of trees), allowing a clear distinction between vegetated and non-vegetated surface types. While the number of patches used can be as high as 19, here we activate three patches, as a compromise in order to avoid increasing too much the computational cost and storage burden of the land surface modelling within the CNRM-AROME modelling system.

**GFLUX**

The "GFLUX" modification consists in reducing the heat flux between the soil surface and the base of the snow. It is designed to reduce the unrealistic soil-snow conduction heat flux due to the unrealistic assumption of an identical soil physical state between the snow-covered and uncovered fractions of the patch.For this configuration, the thermal conductivity of the interface between the lowermost snow layer and the uppermost soil layer, calculated as the harmonic average of the conductivity of each layer, is capped at 5% of its value below a snow fraction of 75%, increasing linearly to reaching its base value when the snow fraction reaches 100%.





The thermal conductivity of the soil-snow interface $Csng$ is thus computed as:

$$Csng = Csng \times \max(0.05, 3.8P_{sn} - 2.8) \qquad (16)$$

**WSN-1**

The modification "WSN-1" consists of adjusting the parameter governing the estimate of the snow cover fraction on vegetation. In this case, the value of $W_{sn}$ in the formula for snow fraction on vegetation (see eq. 5) is lowered from 5 to 1. This modification increases the sensitivity of snow fraction to snow depth, allowing it to reach higher values even with moderate amounts of

snow. The motivation for this modification is similar to that explained for the "GFLUX" modification, but achieved through the reduction of the range of snow depth values with intermediate snow-covered fraction values.

**SG-LSOC**

The modification "SG-LSOC" refers to the use of the SoilGrids v2.0 database (Poggio et al., 2021) for soil textures (proportion of sand and clay), and the activation of the soil organic carbon parameterization effect (Decharme et al., 2016) on soil heat and

mass exchanges. The use of SoilGrids v2.0 is motivated by its better estimate of the soil organic carbon stock than the HWSD database (Batjes, 2016) over France and boreal regions (Tifafi et al., 2018).

**Experimental design**

All these modifications have been defined and tested iteratively, with the aim of improving the seasonal dynamic of snow cover in CNRM-AROME simulations over the European Alps. Only one "major" modification was done per experiment, with

the aim of moving towards a more physical configuration, and/or resolving remaining issues, without overtuning because the land surface model is not the only cause of errors in regional climate modelling. Modifications that consistently reduced discrepancies with the observations used as a reference were retained for the following experiment, reaching at an optimum configuration for simulating snowpack in the European Alps. For clarity and brevity, only the three configurations described above are shown in the main figures of the article. Further results with intermediate configurations are provided in the Appendix.

**2.4  Geographical domain and simulations setup**

The CNRM-AROME simulations were performed at 2.5 km horizontal resolution over a domain that covers the alpine ridge, shown on Figure 2, namely the ALP-3 domain, that is the domain use for the CORDEX FPS Convection (Coppola et al., 2020; Ban et al., 2021; Pichelli et al., 2021; Caillaud et al., 2021; Monteiro et al., 2022). The black highlighted contour of the map defines the mountainous region of the Alps in which all the analyses were performed and corresponds to the

boundaries of the Alpine Convention domain (Convention, 2020). Simulations were driven by atmospheric fields directly from the ERA5 (Hersbach et al., 2020) reanalysis at 50 km horizontal resolution each hour, thanks to the increasing resolution of global reanalyses. This is the first time using the CNRM-AROME climate model, without the need for an intermediate RCM to downscale ERA5 fields. The simulations cover the three-year time period from 01/01/2018 to 31/12/2020, for which we had



the largest number of available observations. The CNRM-AROME atmosphere was initialized using ERA5 fields interpolated
on the ALP-3 domain, and the surface fields were initialized by realizing one time step with these interpolated atmospherical
fields, both on 01/01/2018. A dedicated study was carried out to analyze the impact of the absence of spin-up on the simulation
results, see below Section 2.5. Note that the simulation results were evaluated over the seasons 2018-2019 and 2019-2020, i.e.
from September to August of the following years. The first eight months (and last four months) of the coupled simulations
were therefore not used for the evaluation.

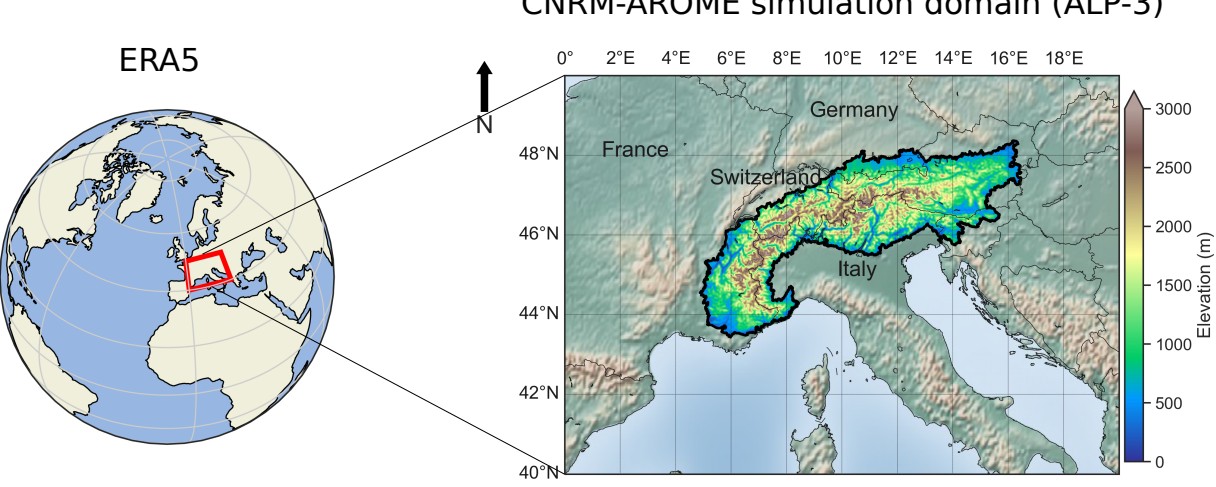

**Figure 2.** Simulation setup displaying the whole domain of simulation, and the contour of the Alpine Convention outline of the Alps, within
which the evaluation was carried out, with the orography of CNRM-AROME at 2.5 km horizontal resolution.

## 2.5  Impact of the initialization (spin-up) approach on snowpack simulations

The high computational cost of coupled surface-atmosphere simulations at 2.5 km horizontal resolution over the whole alpine
ridge prevented us to perform a fully-fledged coupled spin-up for all the configurations. We nevertheless assessed the impact
of a spin-up that is insufficiently long to obtain a balanced ground state. Indeed, albeit soil heat and water content are known
to have a relaxation time ranging from few years to a decade Christensen (1999); Cosgrove et al. (2003), performing our
experiments in transient regime for the soil only affects marginally the results of our study. As reminded by Jerez et al. (2020),
the impact of the spin-up is largely related to the goal of the study (i.e. variables of interest, magnitude of the investigated
changes in a comparison...). In our case, we find that the order of magnitude of the changes of the surface configurations
we performed are way larger than the impact of an unbalanced deep soil heat and water content over snowpack simulations.
Appendix A provides comparisons of model runs using the ES-DIF-OPT configuration using either the default initialization
procedure (described above) or an initialization obtained from 13 years of standalone (offline) simulations of the surface
model from 01/01/2006 to 01/01/2018, driven by near-surface atmospheric fields from the CNRM-AROME coupled model
run driven by the ERA-Interim/ALADIN model pair (Caillaud et al., 2021; Monteiro et al., 2022). Figure A1 in appendix A





confirms that, at the initialization date (i.e. 01/01/2018), the default initialization strongly underestimates snow amounts and provides significantly too wet and warm soil conditions over most of the European Alps, compared to the result of multiple

years of offline simulations. Nonetheless, the snow depth time series at four sites Figure A2 in appendix A show that, after the first 6 months, the effect of the spin-up is negligible compared with respect to the objectives of our study.

## 2.6  Observational references

Various observational datasets taken as reference are used to analyze different aspects of the impact of the choice of the land surface model configurations on simulated snow and atmospheric surface variables and are described in the following

subsections.

### 2.6.1  In-situ snow depth observations

A set of daily in situ snow depth observations is employed to perform an extensive point-scale evaluation of the simulated snow depth values over the 2018-2019 season (i.e. from 01/09/2018 to 31/08/2019) as it was the season for which the largest number of observations was available based on existing consolidated datasets described in Matiu et al. (2021a). Figure 3 shows

the location of the in-situ measurements and their distribution with respect to elevation, spanning the whole alpine ridge over elevations ranging from 0 m to 3000 m.

The observational time series used in this study are quality-checked. The greatest part of the data set was gathered and described by Matiu et al. (2021a), to which we added Austrian stations from the Hydrographic Central Office of Austria (HZB) and GeoSphereAT (i.e. the TAWES and SNOWPAT datasets) and Swiss stations (i.e. the IMIS datasets, (Measurement and

IMIS, 2023)) from the WSL Institute for Snow and Avalanche Research (SLF).

From this large set of in-situ snow depth measurements (i.e. 1005 stations), 266 stations were selected, according to the following criteria :

- The AROME grid point that includes it has less than 150 m difference with the station elevation.

- The AROME grid point that includes it is filled by less than 75% of the high vegetation cover type.

These criteria limit some of the representativeness issues of a point-scale comparison between a local in-situ station and a model grid point representative of a square of $2.5 \times 2.5$ km$^2$ in mountainous regions.





## Locations of the in situ snow depth measurements

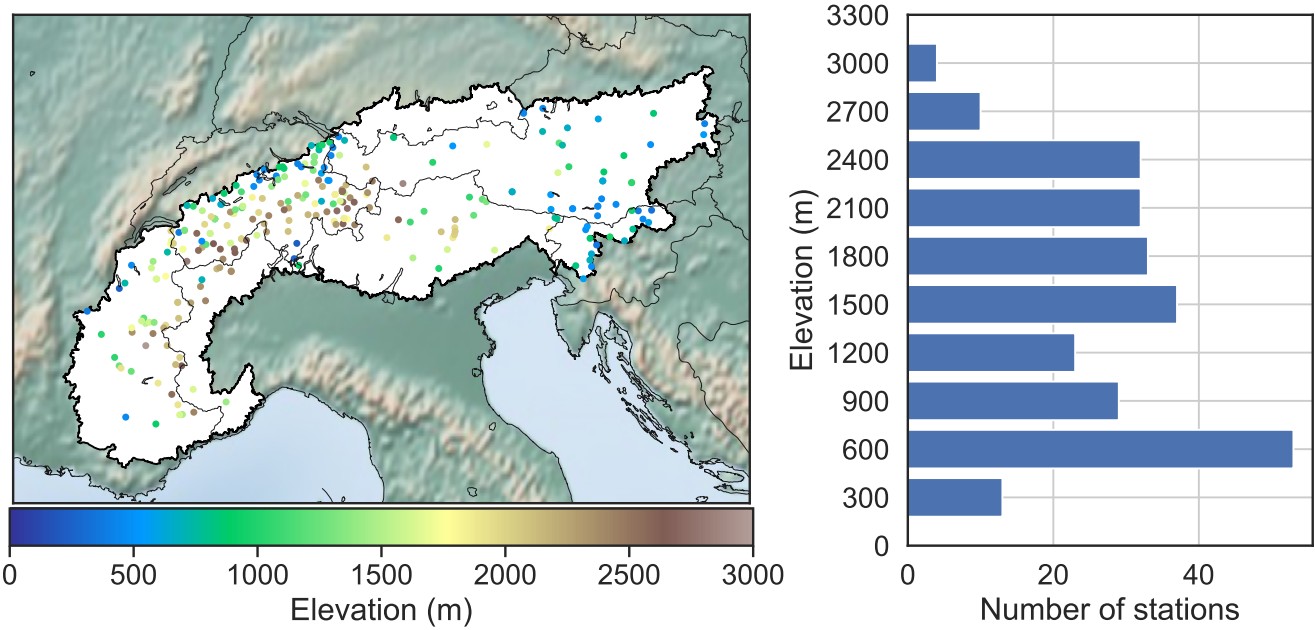

**Figure 3.** Location of the in-situ snow depth observations with their associated number per bins of 300 m width elevation bands.

### 2.6.2 Satellite (MODIS) snow cover duration

A large scale evaluation of the snow cover duration (SCD, defined as the longest consecutive period with snow on the ground based on hydrological years from September to August) was performed using the MODIS/Terra daily normalized difference snow index (NSDI) fields at 500 m for the 2018-2019 and 2019-2020 seasons. These data from the MODIS/Terra sensor have been treated by the National Snow and Ice Data Center (NSIDC) (Hall and Riggs., 2020) and correspond to a daily gap-filled product using an algorithm described in Hall et al. (2010). In this study, MODIS NDSI data were regridded to match the CNRM-AROME horizontal resolutions of 2.5 km using a first-order conservative method.

Figure 4 shows the SCD over the 2018-2019 and 2019-2020 seasons from MODIS over our area of interest regridded at 2.5 km. The MODIS SCD is calculated upon the MODIS NDSI by converting it to a series of binary snow cover maps (absence or presence of snow) using a threshold value NDSI > 0.2. This threshold corresponds to a snow cover fraction of approximately 30% (Salomonson and Appel, 2004). In this study, the CNRM-AROME SCD was computed using snow depth values with a threshold set at 1 cm, motivated by the minimization of error metrics as described in Monteiro and Morin (2023).



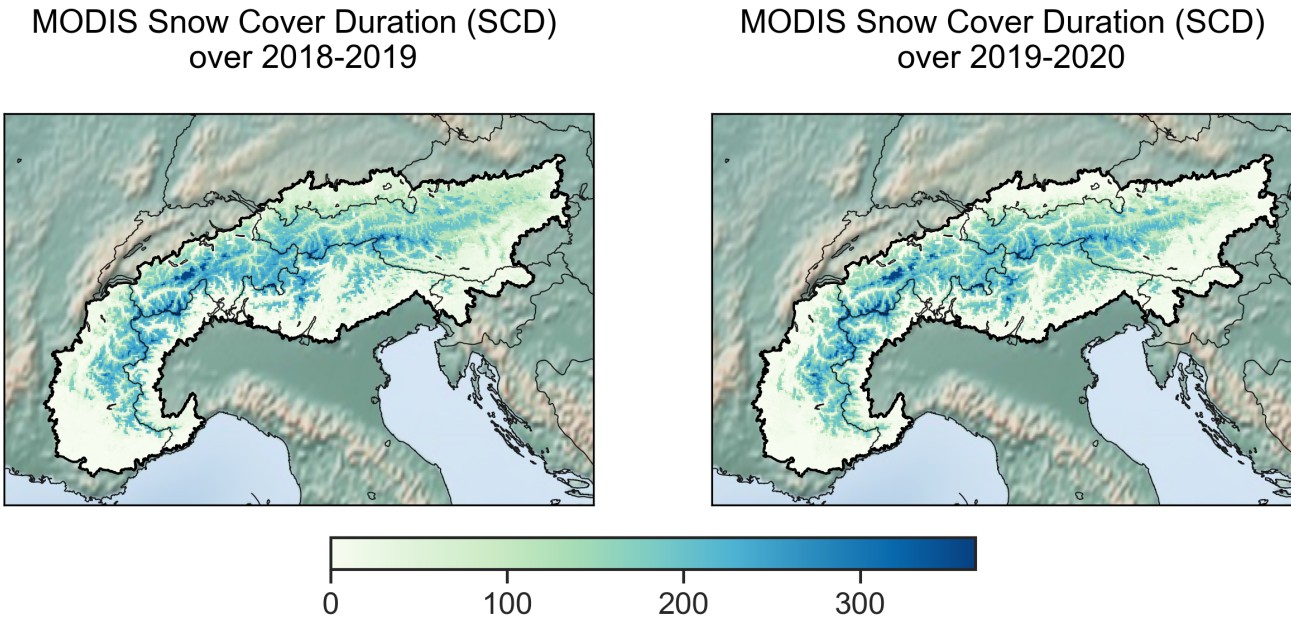

**Figure 4.** MODIS Snow Cover Duration (SCD) over 2 seasons (2018-2019 and 2019-2020) within the contour of the Alpine convention, regridded at 2.5 km horizontal resolution over the CNRM-AROME mesh grid.

### 2.7 Point-scale comparison, elevation bands analyses and used statistics

#### 2.7.1 Point-scale comparison

Sections 3.1 and appendix C introduce point-scale comparisons between individual station measurements and the corresponding CNRM-AROME grid cell. It means that each station is compared to the CNRM-AROME grid cell representative of a $2.5 \times 2.5\,km^2$ square including the station location based on its geographical coordinates.

#### 2.7.2 Elevation bands analyses

Sections 3.1 and 3.2 introduce analyses performed using an elevation-based categorization. Here, we used 300 m-width elevation bands, meaning that for a given elevation band at median elevation **z**, all stations or grid points with an elevation ranging between **z**±150 m are gathered and used. This choice is a trade-off between the heterogeneity within an elevation band, and the inclusion of a maximum of grid points or observations within.

For clarity and brevity, only results for four elevation bands are presented in the main article figures, representing distinct environments : 900 m±150 m for the valleys and low elevation hills, 1500 m±150 m and 2100 m±150 m for the intermediate elevation and 2700 m±150 m for the high mountain conditions. The results for the other elevation bands, not shown, are consistent with the main patterns observed across the analyzed elevation bands.



### 2.7.3 Surface type analyses

The evaluation of the snow cover duration using MODIS remote sensing data is complemented with a categorical analysis by
surface type. Figure 5 show the location and the elevational distribution of points per prevailing surface type (i.e. meaning that
the surface type represents more than 75% of the cover, otherwise is it classified as "mixed") for the CNRM-AROME mesh
grid. The classification per vegetation type is based on the ECOCLIMAP land-use database, from which the 19 vegetations
types have been gathered into three categories : "No vegetation" (i.e. bare ground, rock, permanent snow and ice), "High
vegetation" (i.e. grouping all types of high trees) and "Low vegetation" (i.e. crops, grasslands and shrubs).

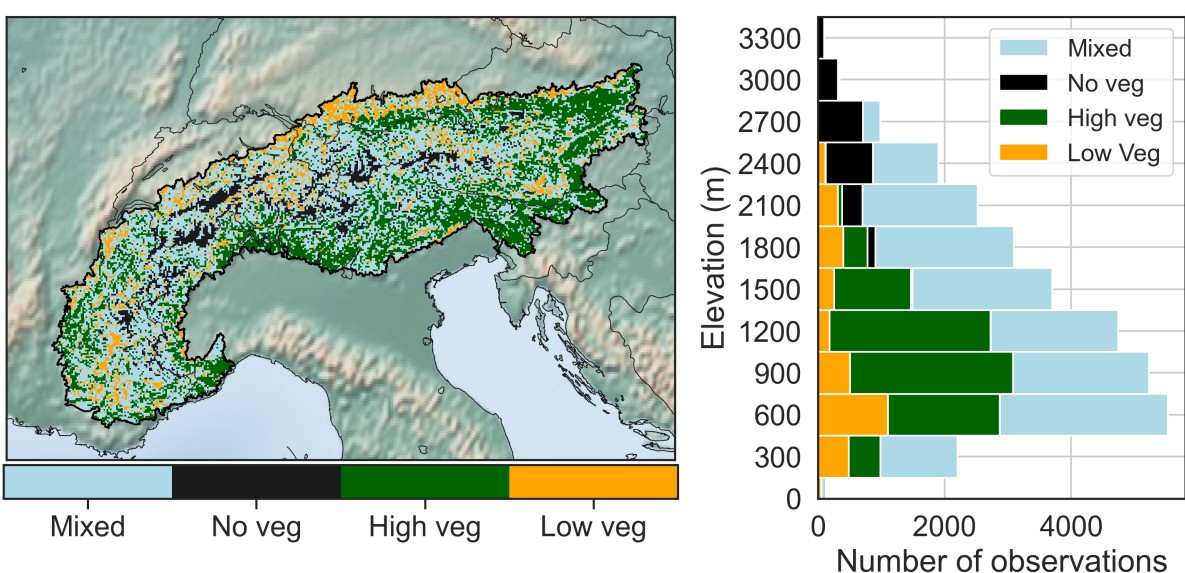

**Figure 5.** Location of the prevailing surface type (i.e. if the surface type represents more than 75% of the cover, otherwise is it classified
as "mixed") for the CNRM-AROME mesh grid within the contour of the Alpine convention. The horizontal barplot on the right shows the
number of grid points per bins of 300 m width elevation bands, classified by prevailing surface type.

### 2.7.4 Statistics

The error metrics used are defined as follows :

– Mean error (ME) : $ME = \frac{\sum_{i=1}^{N}(x_i-y_i)}{N}$

– Correlation (Pearson linear correlation) : $r_{xy} = \frac{\sum x_i y_i - N\bar{x}\bar{y}}{\sqrt{(\sum x_i^2 - N\bar{x}^2)}\sqrt{(\sum y_i^2 - N\bar{y}^2)}}$

with $x_i$ and $y_i$, data $x$ and $y$ at time $i$, $\bar{x}$ and $\bar{y}$ the mean of $x$ and $y$, $\sigma_x$ et $\sigma_y$ respectively the standard deviation of $x$ and $y$ and
$N$ the sample size.



## 3  Results

The presentation of the results is first performed comparing simulation results with a large sample of in-situ snow depth measurements covering the European Alps during the 2018-2019 season. We then evaluate the simulation results in terms of snow cover duration compared to remotely-sensed (MODIS) observations for the two winters 2018-2019 and 2019-2020.

### 3.1  Point-scale evaluation of snow depth values over the 2018-2019 winter

Figure 6 shows the multi-stations mean time series of the height of snow of the three land surface configurations and the reference observations for multiple elevations bands of 300 m width ranging from 900 m±150 m to 2700 m±150 m over the 2018-2019 winter. For each of the elevation bands and configurations the mean errors ($ME$) and the Pearson correlation ($R^2$) calculated over multi-station mean time series are displayed.

The D95-3L configuration simulates snowpack evolution with similar behaviours for all elevation bands. During the accumulation period and until the observed annual snow depth maximum, the simulated multi-stations average values remain close to the observations, with highly correlated variations. Nevertheless, after the observed annual snow depth maximum, deviations from the observed multi-station average values widen, mainly due to the underestimation of melt events (i.e. their frequency and amplitude) in the simulations compared to observations, resulting in less correlated snow depth time series. This leads to a

significantly delayed and higher annual snow depth maximum (i.e. up to two months at intermediate and high elevations) and, to a lesser extent, a delayed end of snow season (i.e. from a few days to a month, partly compensated by faster melting) in the simulation. As stated in introduction, these overestimations of the annual snow depth maximum in its value and its timing are in line with previous studies working with CNRM-AROME climate simulations (Monteiro et al., 2022; Lucas-Picher et al., 2023; Monteiro and Morin, 2023). Overall, the simulated multi-stations average values show high Pearson correlation scores

($R^2$) for the whole snow season, from 0.81 to 0.97, but mean errors ($ME$) can be large, from 2 cm at 900 m reaching 30 cm at 2700 m (i.e. around 10-15% of the annual snow depth maximum).

The snow depth values simulated using the ES-DIF configuration in Figure 6 exhibit large differences with the D95-3L (i.e., the default configuration). During the accumulation period and until the date of the annual snow depth maximum, the simulated multi-stations average values follow similar variations than the observations, slightly underestimating the amount

of snow at 2100 m and below, slightly overestimating it above. Compared to the D95-3L simulated snow depth, melt events appeared to be better captured (i.e. negative variations of the snow depth are better correlated) all along the snow season but often overestimated, notably below and at 2100 m. As a consequence, even if the timing of the simulated annual snow depth maximum often matches with the observation, its value is strongly underestimated at these elevations (from 20 cm at 900 m to 50 cm at 1500 m and 2100 m), and the time of snow disappearance is too early, from 15 days to a month. The ES-DIF $R^2$

value against in-situ snow depth observations is not systematically improved and the $ME$ scores are degraded at all elevations, except at 2700 m. Indeed, the $R^2$ values only increase significantly (i.e. by more than 0.1) at 1500 m compared to the default configuration, and the $ME$ ranges from 2 cm in the D95-3L configuration to -3 cm in the ES-DIF configuration at 900 m, from



14 cm to -15 cm at 1500 m and from 19 cm to -21 cm at 2100 m. This shows that simply using a more complex soil or snow scheme does not warrant improved results compared to a coarser snow or soil model.

The ES-DIF-OPT configuration provides the best estimation of the snow depth values, against in-situ observations. All along the snow season, as shown on Figure 6, its variations are almost identical to the ES-DIF snowpack simulation and thus similar to the observed multi-stations average value until the annual snow depth maximum. Its similar variations of snow depth with ES-DIF is coherent as both configurations share the same physical basis, but the specific features of the ES-DIF-OPT option seem to attenuate the sensitivity to the melt events, leading to simulated snow depth values closer to the observations during

the main melt period after the annual snow depth maximum. Apart from these improvements over the two other configurations, the melt-out date at intermediate elevations (i.e. 1500 m and 2100 m) is still too early in the simulations, as a result of an overestimated melt notably during springtime. Overall, at all elevations, the $R^2$ and the $ME$ scores are improved compared to the other configurations tested. Correlation scores reach very high values with the lowest values of 0.94 at 2100 m, above 0.98 elsewhere. $ME$ values are also strongly reduced compared to other configurations tested, with values changing from -3 cm to

-0.8 cm at 900 m, from -15 cm to -5 cm at 1500 m, from -20 cm to -8 cm at 2100 m and from -8 cm to -5 cm at 2700 m, between ES-DIF and ES-DIF-OPT, respectively.





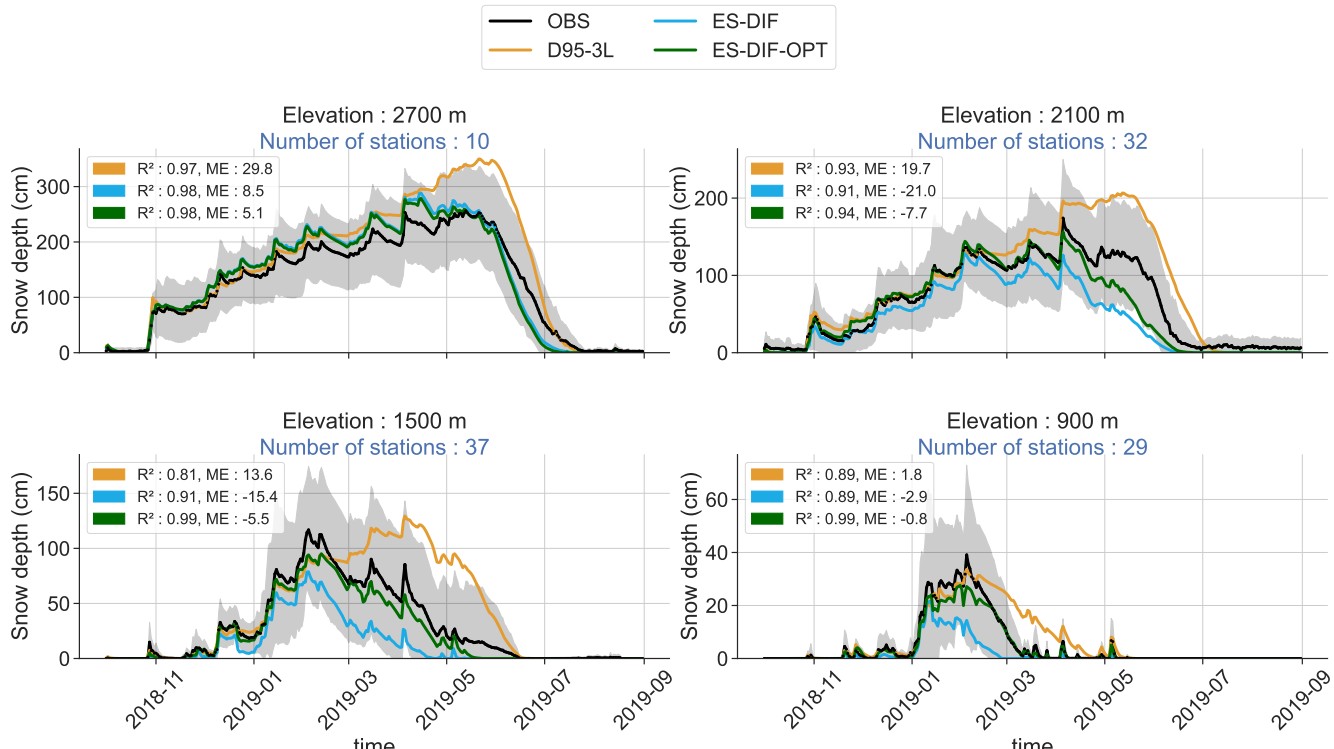

**Figure 6.** Multi-stations average time series of snow depth values for the 2018-2019 winter for the four elevation bands centered at 900 m±150 m, 1500 m±150 m, 2100 m±150 m and 2700 m±150 m above sea level. Colored continuous lines correspond to the simulated multi-stations average time series for each of the configuration : D95-3L in orange, ES-DIF in blue and ES-DIF-OPT in green. Black circle markers correspond to the multi-stations mean time series of the in-situ measurements with the inter-stations standard deviation represented in gray shaded areas. For each elevation bands, the number of stations used to compute the mean and the standard deviation are displayed in blue font. At each elevation bands and for all configurations, the correlation ($R^2$) and the mean error ($ME$) computed using the multi-station time series between the simulated and the in-situ measurements are displayed.

## 3.2 Snow cover duration evaluation using MODIS remote sensing data

In this section, we compare the simulated snow cover duration against MODIS remote sensing data as a reference. The snow cover duration of the two seasons used (2018-2019 and 2019-2020) is averaged for the analysis. Individual seasons show
similar differences between each simulation and the reference. Figure 7 shows the differences in terms of the mean snow cover duration (SCD) over two seasons (i.e. 2018-2019 and 2019-2020) between each experiments and the MODIS SCD, in the European Alps.

Differences obtained using the D95-3L configuration (Figure 7a) indicated a widespread overestimation compared to the MODIS SCD. Apart from a few small areas of underestimated SCD, evenly distributed over the Alpine ridge, no specific
region seems to show more marked differences than others. Conversely, the SCD of the ES-DIF configuration (Figure 7a) is





largely underestimated, with an amplitude that appears to depend on elevation. Indeed, stronger negative values of the difference between simulated and observed SCD (Δ SCD) are found on the outer edge and in the northeastern part of the European Alps while a few patch of slightly positive values of Δ SCD are located along the ridge. The last configuration ES-DIF-OPT provides the best match with observed SCD values, reducing the magnitude of the differences with MODIS SCD at each location of the
study area compared to the D95-3L and ES-DIF configurations. The northeastern part of the Alps concentrates more areas with an underestimation of the SCD, while the rest of the Alps presents a widespread slight overestimation.

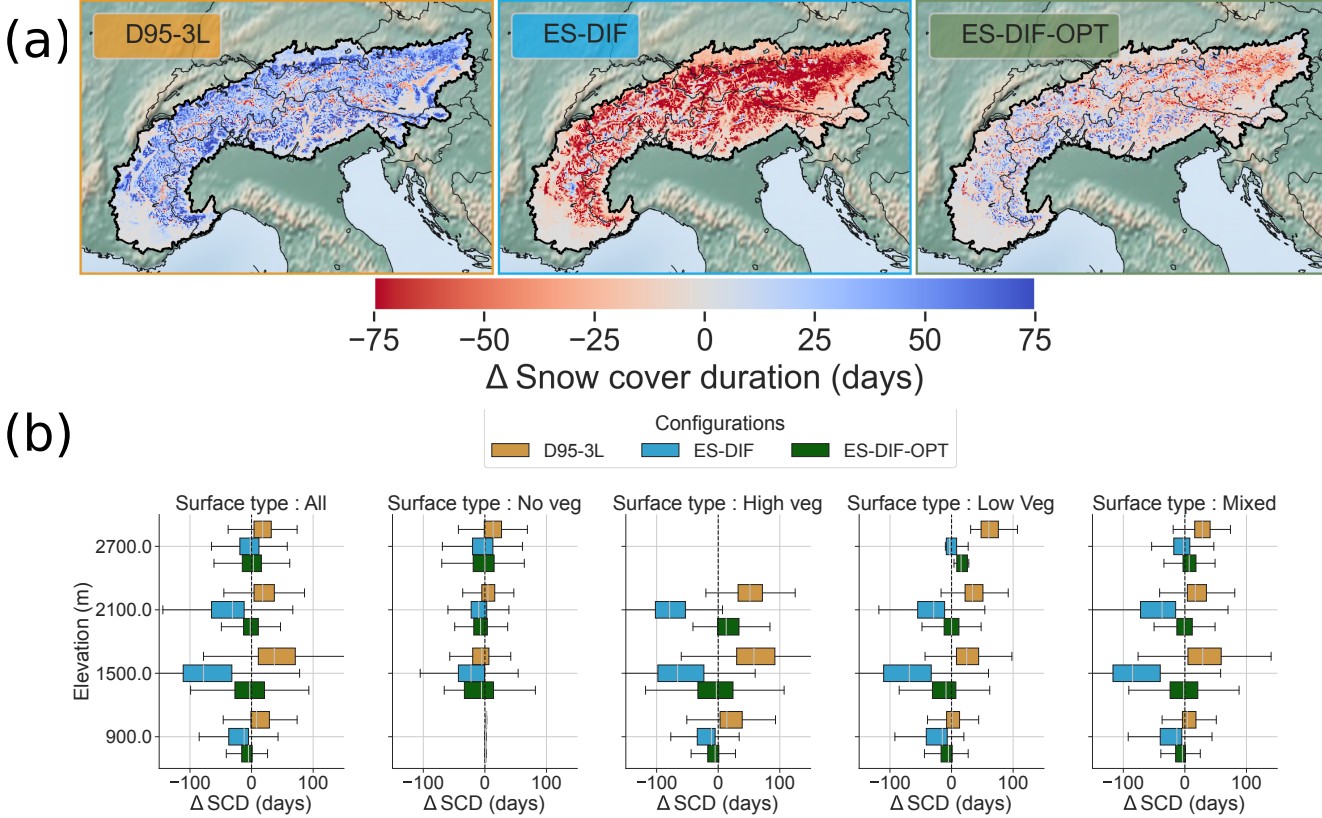

**Figure 7.** Snow cover duration differences between the simulation results using the different configurations and MODIS observations in the European Alps. Note that MODIS products initially at 500 m horizontal resolution have been regridded over CNRM-AROME horizontal resolution (2.5 km) grid using a first-order conservative method. **(a)** Map of the average differences (mean error) of the snow cover duration (SCD) over 2 seasons (2018-2019 and 2019-2020) for each configurations compared to MODIS SCD. **(b)** Boxplot representing the spatial distribution of the average differences (mean error) of the SCD over 2 seasons (2018-2019 and 2019-2020) compared to MODIS SCD for each datasets for the six elevation bands 900 m±150 m, 1500 m±150 m, 1800 m±150 m, 2100 m±150 m, 2400 m±150 m and 2700 m±150 m above sea level. Each column corresponds to the values classified by prevailing type of surface (see section 2.6.2 for details).

The elevational distribution of the differences are represented with boxplots on Figure 7b, categorized by prevailing type of surface based on the ECOCLIMAP I land surface classification, used as a physiographic database in the CNRM-AROME





simulations (see section 2.2.1). It informs us further on the specific behaviour of the simulated behavior of the snow cover
regarding the elevation and the surface type.

Looking at the "All" category (i.e. gathering all grid points regardless of their surface type) in Figure 7b confirms quanti-
tatively what was found on the map Figure 7a. The results from the D95-3L and ES-DIF experiments exhibit distributions of
SCD differences strongly biased towards an overestimation and an underestimation, respectively, for all elevation bands. The
median SCD differences of D95-3L range from +5 days to +40 days, and the ES-DIF from -5 days to -75 days, both larger at
1500 m elevations than above and below (i.e. in terms of median values and larger in terms of variance). The ES-DIF-OPT
configuration is at the center of the two other configurations, showing zero-centered distributions of SCD differences, with
greatly reduced variance for most elevation bands. Note that the "mixed" surface type (i.e. all grid cells with less than 75%
prevalence in each category) shows similar distributions of differences that the "All" categories that gathers all grid points.

While an analysis by elevation alone would lead us to interpret that greater variations can be found at intermediate elevations,
categorization by dominant surface type (i.e. "No veg", "High veg" and "Low veg") brings more contrasting results, supporting
the hypothesis that the simulation results mainly depend on surface types.

It is on the prevailing "No veg" surface type that the simulated SCD values show the smallest differences with the MODIS
reference values, as well as between the different experiments, meaning that the changing surface configuration has only a
marginal effect on the simulation of the SCD for this surface type. Indeed, the median $\Delta$ SCD values of all experiments
combined ranges from -15 days to +10 days at most, with this value increasing slightly with elevations for each experiment,
with only slight improvements in the scores provided by the ES-DIF-OPT configuration.

The highest $\Delta$ SCD and the most contrasted behaviors between experiment, results are found in the "High veg" surface type.
On this type of surface, the D95-3L experiment strongly overestimates the SCD, with median $\Delta$ SCD values of +10 days at 900
meters, increasing to +50 days at 1500 m and 2100 m. The ES-DIF experiment shows the opposite behaviour, with a median $\Delta$
SCD of -5 days at 900 m, increasing from -60 days to -70 days at 1500 m and 2100 m respectively. For the "High veg" category,
the ES-DIF-OPT configuration brings substantial improvements in scores, with a median difference of the SCD with MODIS
of -3 days at 900 m, -1 day at 1500 m and +10 days at 2100 m.

For the "Low veg" surface type, the simulated SCD values using the D95-3L configuration are overestimated above 900 m
compared to observations, with increasing differences with MODIS SCD values, reaching +60 days at high elevations. The
ES-DIF $\Delta$ SCD values are negative at all elevations, but exhibit higher discrepancies at intermediate elevations (i.e. at 1500 m
and 2100 m) with median values between -30 days and -60 days. Again, the ES-DIF-OPT configuration shows the smallest
differences, with median values comprised in the -10 to +15 days range.

Overall, the analysis of differences between simulated and observed snow cover duration values ($\Delta$ SCD) demonstrate a
clear added value of the ES-DIF-OPT configuration, reducing discrepancies across all surface types and elevations. Indeed,
it often provides zero-centered median values of the differences, as well as a smaller standard deviation of the differences
than the other experiments. Analyses by elevation band show that differences are often larger in terms of median and standard
deviation at intermediate altitudes (i.e. 1500 m and 2100 m), which may be linked to partial or intermittent snow conditions,
which are more sensitive to atmospheric and ground physical state, and therefore more difficult to model adequately. A closer



look at each type of surface also shows that the main differences between our experiments lie in the presence of vegetation,
and is higher for "High veg" than for "Low veg" surface types.

## 4 Discussion

In this study, we analyzed the results of various surface configurations in the CNRM-AROME high-resolution regional climate model on snowpack simulations in the European Alps, tested through coupled model simulations at 2.5 km horizontal resolution, driven by the ERA5 large-scale reanalysis.

Various reference datasets and indicators were used to evaluate multiple aspects of the snowpack simulations, including the snow cover duration using remote sensing data from MODIS, a multivariate analysis at four well-instrumented sites (including air temperature and radiation balance terms), and a comparison of the snow depth on a large set of in-situ stations covering the Alpine ridge.

These comparisons allowed us to gain insight into the challenges of snowpack simulation within the different AROME sur-
face configurations and ultimately document an optimized land surface configuration. In the subsequent sections, we examine the various causes for the successful and unsuccessful modifications that we tested, remaining problems and limitations, and finally propose perspectives to further improve snow simulation in the European Alps and beyond, using CNRM-AROME or other regional climate models.

### 4.1 D95-3L : an overly simplistic configuration failing at reproducing the seasonal snowpack evolution

The default surface configuration, namely D95-3L in this study, is based on a force-restore approach for the soil exchanges and a single-layer parameterization for snow. This surface configuration, used in recent climate study frameworks such as the CORDEX FPS on Convection (Coppola et al., 2020; Pichelli et al., 2021; Ban et al., 2021; Caillaud et al., 2021) and in Météo-France's numerical weather prediction system using AROME (Seity et al., 2011; Brousseau et al., 2016) exhibits a series of issues over mountainous regions, such as a cold bias at high elevations (Vionnet et al., 2016; Monteiro et al., 2022; Arnould
et al., 2021; Gouttevin et al., 2023), and a generalized overestimation of the amount of snow (Monteiro et al., 2022; Monteiro and Morin, 2023).

These problems have been reproduced in our experiments using two years of regional simulation driven by ERA5, and our multiple comparisons allow us to characterize them further.

In section 3.1, in particular on Figure 6, we demonstrated on a large set of in-situ snow depth observations that this con-
figuration is unable to provide a satisfactory seasonal evolution of the snowpack at all elevations, from the lowest studied at 900 m to the highest at 2700 m. While the first accumulations are generally well correlated with observations, the start of the melt period, from late winter at low elevations to late spring at high elevations, marks the beginning of strong divergences with observations. From this point onwards, the magnitude of melting events is often severely underestimated, or even completely missed, leading to a delay and overestimation of the snow depth annual maximum, and then of the end of the snow season,
which can last up to a month.



Section 3.2 confirms the underestimated magnitude of snow melt at a larger spatial scale, displaying a generalized overestimation of the duration of snow cover at all elevations on the map and boxplots Figure 7. Nevertheless, the categorization of differences by surface type (i.e. "No vegetation", "Low vegetation" and "High vegetation") nuances this analysis. The overestimation appears to be particularly linked to the presence of vegetation, as the "No vegetation" category shows close
to zero-centered differences compared with the reference, while the strongest overestimation cases are found for the "High vegetation" category above 1500 m and the "Low vegetation" category above 2100 m.

Multivariate analyses carried out on four well-instrumented sites and provided in appendix C reinforce these findings. Indeed, on Figure C2a the three sites characterized by mixed surface types (i.e. Davos, Torgnon and Weissfluhjoch, that include vegetated surface types, see table C1) show a clear de-correlation of snow depth variations after the date of the annual snow
depth maximum resulting in a strongly delayed melt-out date in the case of Torgnon and Davos.

Based on these evidences, multiple statements can be formulated to explain the widespread overestimation of snow in the simulation using the D95-3L configuration. The overestimation of winter snowfall at high elevation, already reported in past studies (Monteiro et al., 2022; Lucas-Picher et al., 2023; Monteiro and Morin, 2023) may contribute to provide overestimated snow accumulation at the highest elevation bands. Nonetheless, a significant part of this overestimation is likely to be attributed
to the design of the configuration itself, and more specifically how melt is computed in the presence of vegetation, explaining its propensity to make larger errors over these surface types. As stated in section 2.3.1, in the presence of vegetation, the calculation of the melting temperature becomes composite between the surface temperature and the deep soil layer temperature (see eq. 10 in section 2.3.1). The latter leads to a decoupling between melting intensity and the sub-daily oscillations of the energy balance, which unfortunately results in an underestimation of the snow melt in many cases.

## 4.2   ES-DIF : an intermediate complexity surface configurations holding conceptual issues in coupled surface-atmosphere model if only one patch is used

In order to solve some of the issues of the original, simplified D95-3L configuration, we experimented a more detailed and physically-based land surface configuration, using the multi-layer soil scheme ISBA-DIF (Boone et al., 2000) together with the explicit multi-layer snow scheme ES (Boone and Etchevers, 2001), using only one patch for the NATURE tile in SURFEX.
However, the majority of the results from our study indicate no improvement, and in some cases, a degradation of the simulation of snow depth. Against the multi-stations mean snow depth time series on Figure 6 in section 3.1, the ES-DIF configuration displayed degraded scores of $ME$ and $R^2$ compared to D95-3L, except at the highest elevations studied 2700 m. As opposed to the D95-3L experiment, the simulation using the ES-DIF configuration underestimated the snow depth value from the first melt event after the date of the snow depth annual maximum to the end of the snow season, often anticipated
from 15 days up to a month. Albeit this configuration generally underestimates the amount of snow, we note that it simulates variations of the snow depth (see e.g. on Figure 6) that are much better correlated with the observations than the simulation using the D95-3L configuration. We hypothesize it to its explicit treatment of the snowpack and an enriched description of the physical processes within the snowpack (i.e. liquid water retention, phase change and compaction).




Section 3.2 highlights the underestimation of the snow cover with this configuration, by displaying a spatially generalized
underestimation of the snow cover duration compared to MODIS, except at the highest elevation, as demonstrated by the near-
zero centered differences at 2700 m in the boxplots Figure 7b. As for the D95-3L configuration, the boxplots show that the
differences are enhanced in the presence of vegetation and at intermediate elevations, while the lowest differences are found at
high elevations for the "No vegetation" surface type.

Lastly, the appendix C consolidates the analysis of the relationship between the magnitude of the underestimation and the
elevation and the presence of vegetation. Simulations at the Davos site exhibit a large underestimation of its snow depth
time series (see Figure C2a in appendix C), where the simulations struggle to even provide a continuous snow cover due to
overestimated melting events, leading to total melting of the snow cover even in mid-winter, where observations only show a
partial melt of the snowpack. Similarly, the Torgnon and Weissfluhjoch sites display an overestimated late spring melting rate.
The examination of the radiative balance of sites showing significant underestimation of the snow depth (i.e. Col de Porte and
Davos) reveals a higher than observed amount of radiation absorbed, due to overestimated incoming radiation amounts and
partially snow-covered surfaces.

As for the overestimation of the snowpack simulated by the D95-3L configuration discussed in in section 4.1, several reasons
can be invoked to explain the underestimation of the amount of snow and the exaggerated snow melt intensity at intermediate
elevations and in the presence of vegetation in the ES-DIF configuration. Apart from the biased atmospheric conditions, such
as the irradiance biases illustrated in Figure C2b in appendix C whose effects are discussed further in a separate subsection, we
suspect conceptual choices made in ISBA for computing the surface energy balance in the presence of snow on the ground to
bring about numerous drawbacks on snowpack modeling, exacerbated in the case when only one patch is used.

Figure 8a illustrates in a simplified way the heat exchanges between vegetation, soil surface and atmosphere in the presence
of snow on the ground in a forested area, and Figure 8b how it is modeled using the ES-DIF configuration.

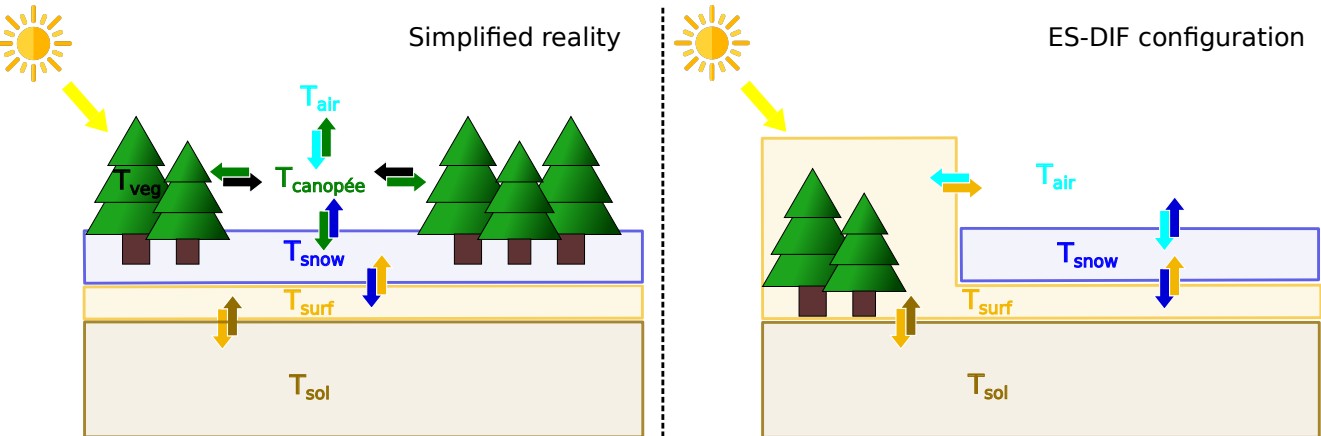

**Figure 8.** Schematic illustration of the heat exchanges between the different component of a forested area in the presence of snow for two
cases : a simplified case to represent the "reality" and the way the ES-DIF configuration represents it.



What we would expect to observe in forested areas as well as in open areas, is that even a small amount of snow covers most of the surface even if some of it is intercepted by trees, and tree trunks and branches remain uncovered. Consequently, the soil surface would be isolated from the canopy and the atmosphere and would interact mainly with the overlying snow and underlying soil by thermal diffusion and latent heat of phase change. Just above the soil surface, snowpack exchanges energy with a part of the atmosphere strongly influenced by the presence of the canopy, reducing momentum, shadowing part

of the incoming shortwave, but emitting longwave radiation and latent heat flux through evapotranspiration. In this case, the air temperature above the canopy $T_{air}$ would be influenced by the surface components mainly through turbulent fluxes, strongly modified by the canopy roughness.

        In ISBA, however, the increase in snow amount on the ground is accounted for by progressively covering the surface and modifying surface variables such as the roughness length $z_0$ and albedo $\alpha_s$, modulating the amplitude of the radiations absorbed

and the turbulent fluxes (Boone and Etchevers, 2001). However, in forested areas, as the vegetation is included in a composite soil-vegetation surface layer, this means that snow gradually "replaces" vegetation.

        To ensure that the effects of the vegetation on surface variables such as lowering albedo and increasing roughness length are adequately represented, notably to preserve turbulent fluxes to the atmosphere, the parameterization of the snow fraction over vegetation (see eq. 5) has been developed in such a way that part of the high vegetation surface remains uncovered even when

a large amount of snow is present.

        Unfortunately, since a single surface temperature is used for both snow-covered and snow-free surface, this approach inhibits most of the insulating effect of snow cover on the underlying ground.

        Napoly et al. (2020) and Nousu et al. (2023) have documented the effects of this approach on the simulated snowpack in forests and examined its effects on the radiation balance and turbulent fluxes using standalone simulation of the surface,

forced by observed atmospheric variables. The authors found that the model significantly underestimate the snow extent and depth at several forested sites. The main reason is an unrealistic coupling between the soil surface and the atmosphere in the case of a partially snow-covered surface, which leads to a strong overestimation of the diurnal amplitude of the soil heat flux. During daytime, a considerable amount of energy can be absorbed over the snow-free area (mainly through incoming shortwave radiation), overestimating the warming of the surface layer below the snow-covered areas, heating the snowpack

from below and likely causing excessive melting at its base. They also note that this approach tends to overestimate the latent heat fluxes, mainly due to the overestimation of soil evaporation, and that the strong coupling also leads to unrealistic surface cooling during nighttime, highlighted by an average cold bias of soil temperatures on the order of -5°C.

        Initially intended as a compromise between the simulation of turbulent fluxes in the presence of vegetation and the insulating effect of the snowpack, this way of treating snow over in vegetated areas turns out to be largely unbalanced, to the detriment

of soil temperature and snowpack simulation.

        In our study, we suspects these feedbacks to constitute the leading mechanisms causing a large part of the underestimation of snow cover in the ES-DIF configuration, particularly in the presence of vegetation and at intermediate elevations, where we often find partially snow-covered surfaces. We also note that there are little to no cases where the ES-DIF configuration, with only one patch, has been implemented in offline or coupled modelling system.



### 4.3 ES-DIF-OPT : an optimized configuration towards addressing conceptual issues in snow representation in the ISBA LSM

The ES-DIF-OPT simulation uses identical soil and snow schemes as the ES-DIF setup, to which a number of modifications have been added. Their primary aim is the reduction of excessive snow melt in the ES-DIF simulations, discussed in section 4.2.

Section 3.1 demonstrates that the ES-DIF-OPT configuration provides the best seasonal evolution of the snow depth at all elevations (see Figure 6), systematically increasing $R^2$ and decreasing $ME$ values compared to the other two configurations. Its snow depth variations during accumulation and melt periods are similar to the ES-DIF simulation, but the frequency and magnitude of melting events are in much better agreement with observations.

Comparisons with MODIS snow cover duration (section 3.2) show a clear improvement in the seasonality of the snow cover compared to the other configurations. The differences are smaller for all elevation bands and surface types, with a greater reduction on the vegetated surface types and at intermediate elevations, as illustrated in Figure 7, by medians of differences close to zero, and distributions that show a reduced variance compared to the other configurations.

More nuanced results are obtained in the point-scale comparison at four well-instrumented stations, see appendix C. As shown on Figure C2, only the Davos site exhibits a significant improvement in the simulated snow depth with a reduced $ME$, while at the other sites the differences with the ES-DIF and ES-DIF-OPT configurations are marginal or negligible as at the Col du Lac Blanc site.

Figures B1 and B2 in appendices B1 and B2 provide a more thorough analysis of the distinct impacts of each modification and clarify the source of the biases in the snowpack simulation identified in the ES-DIF configuration.

The 3-PATCHS modification, as described in section 2.2.1, splits the calculation of energy and mass balances at the subgrid scale, performing an independent calculation for each patch and summing the fluxes obtained, rather than averaging the surface variables of each surface type and performing a single calculation. Its effects are significant on the simulation of the snowpack, as shown on Figure B1, by reducing the $ME$ and increasing the $R^2$, especially at low and intermediate elevations. We note that its impacts are limited above 1500 m, and negligible at 2700 m (i.e. where most of the grid cells are devoid of vegetation, see Figure 2.6.2). The boxplots of the snow cover duration differences in Figure B2 show that the 3-PATCHS modification has limited impact in case of a prevailing (i.e. cover more than 75% of the grid cell) "No veg" surface type, while the reductions of the differences are high for the "Low veg" and "High veg" surface types, and obviously for the "Mixed" surface type at low and intermediate elevations where the proportion of vegetation is high. Thus, the significant improvements brought about by this modification show that in many cases the aggregated characteristics of the surfaces, when using a single patch, cause exaggerated melt. It is likely that in these cases, the resulting aggregated surface characteristics present sufficiently low surface albedo and high roughness length to trigger the undesirable mechanism described in detail in section 4.2. This leads to undue melting either at the base of the snowpack through an over-estimation of heating of the soil surface under snow-covered surface or at its surface through the overestimation of turbulent fluxes. This hypothesis is in line with the greater impact of this modification seen in the presence of high vegetation, where the aggregation of surface characteristics produces a higher



roughness length and a lower albedo, and at intermediate altitudes, where the surface temperature is often near the freezing point. It is also consistent with the low or zero impact on most of the well-instrumented sites, as Col de Porte and Col du Lac Blanc are composed of unique surface type and at Torgnon and Weissfluhjoch sites that contain only a small fraction of "Low vegetation" and are located at high elevation, where the sensitivity of the snowpack simulation to changes in their energy balance is weaker, due to low surface temperatures.

The GFLUX modification consists in lowering the intensity of heat exchange between the surface and the overlying snow layer in the case of partially snow-covered grid cells (see section 2.3.3). This is a pragmatic approach, albeit not grounded on physical principles, to decrease exaggerated melt due to excessive melting at the base of the snowpack in the case of a partially snow-covered surface, as described in section 4.2. Figures B1 and B2 demonstrate significant improvements at the same elevation bands and surface types as the 3-PATCHS modification (i.e. intermediate elevation bands and in the presence of vegetation). The effectiveness of the modification in these areas supports the hypothesis of an overestimation of the ground heat flux, inducing basal melt in partially snow-covered surfaces, when this modification is not implemented.

Compared to the 3-PATCHS and GFLUX, the WSN-1 modification shows only slight improvements, probably limited by the chosen value, which may still be too high to increase substantially the sensitivity of the snow fraction parameterization. However, the chosen value is a first attempt as a compromise to avoid the over-reduction of the turbulent fluxes in the near-surface atmosphere (see section 4.2 for further details).

Overall, the ES-DIF-OPT configuration outperformed the other two in every aspect of the snow simulations investigated in this study. We suggest considering this configuration as a basis for future simulations using the CNRM-AROME model. Although the study concentrates the evaluation on seasonal snow cover in mountainous areas, a clear improvement in the representation of snow events in lowlands is also expected. Snow events in lowland areas, which are typically less intense than those in mountainous locations, are unlikely to cover the entire surface of the grid cells within the ISBA model. Consequently, these events would most likely be underestimated (snow depth, snow cover duration) using the ES-DIF configurations with only one patch, leading to unrealistically early melting, which will be greatly reduced using our optimised configuration. Specific investigations are required to assess theses expected improvements explicitly.

## 4.4 Perspectives regarding error compensations and the effects of land surface/atmosphere coupling

The present work has only addressed a fraction of the sources of model errors, focusing on those related to the surface scheme. The results section, and more specifically the appendix B1, provides evidence of numerous other potential sources of error that can have major implications for the simulation of the snowpack. This justifies that we did not attempt to achieve "perfect" match with observations through modifications of the land surface scheme, both because observations are also affected by uncertainties, and because other sources of errors, in particular in the atmospheric part of the CNRM-AROME model, certainly play a role in the overall performance of the model used.

The site of Torgnon and Col du Lac Blanc on Figure C2a in appendix C, both located above 2000 m above sea level, show an overestimated amount of snow all along the season, with discrepancies progressively widening during the accumulation period. During this period (early in the season), only few melt events occur (in both observations and simulations), and appear to be



well captured by the different model configurations as shown by simulated snow depth variations highly correlated to observations. These findings would therefore point towards an overestimation of the snowfall amount by the CNRM-AROME model

at these high elevation sites. This overestimation of snowfall is consistent with previous studies that used the CNRM-AROME model, such as Monteiro et al. (2022), Monteiro and Morin (2023) and Lucas-Picher et al. (2023), which reported a widespread overestimation of winter precipitation at high elevations over the Alpine ridge. This overestimation is also confirmed by (Haddjeri et al., 2023) who used the NWP AROME precipitation fields to force standalone SURFEX simulations, resulting in an overestimated amount of snow over multiple areas the French Alps. Nonetheless, this overestimation may not be systematic,

as shown at the Weissfluhjoch site, located at 2500 m, presenting snow depth values during the accumulation period close to the observations, and at 2100 m and 2700 m compared to the multi-stations mean on Figure 6 section 3.1.

The simulated irradiance values (incoming longwave and shortwave) at each of the well-instrumented sites display systematic biases (see Figure C2b appendix C), namely an overestimation of the shortwave and an underestimation of the longwave. These can be substantial during the snow season, reaching -35 W m$^{-2}$ for the incoming longwave radiation, and +50 W m$^{-2}$

for the incoming shortwave radiation. They corroborate previous studies, that have documented these biases in the NWP version of AROME (Vionnet et al., 2016; Quéno et al., 2020; Gouttevin et al., 2023) and the CNRM-AROME climate simulation (Lucas-Picher et al., 2023), all attributing it to the underestimation of the cloud cover over mountainous regions. In our study, the positive incoming shortwave biases may play a key role in the exaggerated melt frequency and magnitude in the ES-DIF configuration at low and intermediate elevations. Indeed, in cases of partially snow-covered surfaces, incoming shortwave bi-

ases may strongly favor the warming of the soil surface and enhance the feedback leading to undue melt, described in detail in section 4.2.

Conversely, the negative biases of the incoming longwave radiation are likely to contributes to the underestimated melting rate at some high elevation sites such as the Col du Lac Blanc site, as Lapo et al. (2015) and Quéno et al. (2020) demonstrated that a deficit of incoming longwave radiation often leads to a large underestimation of melt intensity. These biases are also

prone to lead to the over-cooling of the surface during nighttime, increasing the near-surface stability of the atmosphere and triggering the self-sustaining stability feedback energy loss described by Lapo et al. (2015), in which the increasing stability inhibits turbulent exchanges, thus accelerating the surface cooling. As reported by Gouttevin et al. (2023), the biased incoming longwave explains a large fraction of the surface temperature biases in AROME NWP simulations at Col du Lac Blanc, and contributes to the cold bias in near-surface air temperatures as it is diagnosed from it. These biases and associated feedback

are likely to contribute to the cold biases observed at other high elevation sites, such as Torgnon and Weissfluhjoch, and we can even expect it to be relatively widespread across the Alpine ridge, where a generalized winter and spring cold bias at high elevations have been documented in several studies (Monteiro et al., 2022; Lucas-Picher et al., 2023; Monteiro and Morin, 2023).

In the end, in addition to the surface modelling errors, our experiments also corroborate two substantial errors sources in

terms of irradiance values. The literature suggests that the biases have conflicting impacts on the snowpack and are location-specific (Lapo et al., 2015; Quéno et al., 2020). The location factors that significantly influence the effects of the biases include





exposure, elevation, and climate type. Therefore, disentangling the impacts of individual biases in coupled surface-atmosphere simulations is almost impossible.

## 5 Conclusions

In this study, we investigated three-year long simulation results using three main surface configurations of the coupled surface-atmosphere convection-permitting regional model CNRM-AROME over the European Alps at 2.5 km horizontal resolution. It is the first case where detailed investigations using CNRM-AROME as a regional climat model are performed using a land surface configuration strongly deviating from the land surface configuration used by AROME for NWP applications.

By leveraging different datasets used as a reference, we explore multiple aspects of the simulation of the seasonal snow
cover, such as an extensive analyses of the snow depth time series over 2018-2019 and a spatially exhaustive comparison of the snow cover duration using MODIS remote sensing data over 2018-2020.

Based on this analysis, we documented further the issues of the current land surface configuration used in CNRM-AROME for climate studies (Coppola et al., 2020; Caillaud et al., 2021) and numerical weather prediction (Seity et al., 2011; Brousseau et al., 2016) (i.e. D95-3L). We shed lights onto the potentials and limitations of an enriched surface configuration using
intermediate complexity, multi-layer soil (Boone et al., 2000; Decharme et al., 2011) and snow (Boone and Etchevers, 2001) schemes (i.e. ES-DIF). Ultimately, we introduced an optimized land surface configurations based on the ES-DIF configuration (i.e. ES-DIF-OPT).

We confirmed the documented issues of the D95-3L default configuration (Monteiro et al., 2022; Lucas-Picher et al., 2023; Monteiro and Morin, 2023), namely a spatially widespread overestimation of the amount of snow and delay of the end of
the snow season up to month and a half. Using a categorical analyses of the snow cover duration by surface type and a comprehensive comparison of the energy balance at some punctual sites reveal wider discrepancies on vegetated areas, and a clear underestimation of melt during most of the snow season. These issues were mainly attributed to the over-simplicity of the snow scheme, including the snowpack in a soil-vegetation-snow composite layer.

We demonstrated that the multi-layer soil and snow schemes configuration ES-DIF failed at reproducing the seasonality of
the snow cover in the European Alps if only one patch is used. Although the many additional physical processes (compared to the D95-3L configuration) enable this configuration to capture well most of the variations of the snow depth during the snow season, the simulation presents a widespread underestimation of the duration of the snow cover below 2500 m particularly at intermediate elevations and in the presence of vegetation, resulting from an exaggerated melt. We discussed the origin of this issue, already reported in standalone surface simulations by Napoly et al. (2020) and Nousu et al. (2023), and attributed to
conceptual choices in the ISBA LSM with respect to the snow cover fraction. This behaviour results indeed from an underestimated snow cover fraction in vegetated areas, leading to an over-warming of the soil surface below the snowpack provoking undue melt at its base.

This issue is a major topic as it appears in many modeling system using similar configurations. Indeed, it is identified to be causing a wide underestimation of the snow cover in boreal regions in the CNRM-CM6 GCM (Decharme et al., 2019) model



results, and likely explains the early melt in the latest simulations of the CNRM-ALADIN RCM, also using a similar land surface configuration, as shown by Monteiro and Morin (2023).

Finally, we introduced the ES-DIF-OPT configuration, mostly based on existing options in SURFEX but not activated hitherto, which provides the best estimation of the seasonality of the snow cover and daily evolution of the height of snow over a large sample of observations in the European Alps. We find the reduction of the ground heat flux and the splitting of the energy 785 balance for three surface type categories to constitute the major contribution to lowering the errors. Their effectiveness confirms the hypotheses put forward to explain the exaggerated melt in the simulation using the ES-DIF, and enable the simulation of the snowpack to be satisfying regarding the references used in many cases.

At the end, the ES-DIF-OPT configuration consists of an adjusted configuration, which limits the shortcomings of the ES-DIF configuration, offering an interesting alternative that we recommend for future simulations using the CNRM-AROME 790 model and other modeling systems using similar configurations, pending that a more physical solution can be implemented in coupled model configurations. Indeed, the ES-DIF-OPT configuration still fails to adequately capture the evolution of snow beneath the forest and around the mean snowline elevation, where partially snow-covered surfaces are often observed. These results advocate for the use of explicit vegetation modules instead of composite soil-vegetation approach, enabling the current snow-covered fraction parameterisation to be redefined as the below-canopy snow coverage, reducing the excessive sensitivity 795 of simulation results to this very uncertain parameterization (Nousu et al., 2023). Such approach is already implemented in most LSMs (e.g. HTESSEL (Balsamo et al., 2009), NOAH-MP (Niu et al., 2011), CLM5 (Lawrence et al., 2019), JULES (Best et al., 2011)), and recent developments have been made to implemented it in ISBA throught the MEB (Multi Energy Balances) module (Boone et al., 2017; Napoly et al., 2017). Unfortunately, this advanced version of the land surface scheme SURFEX cannot yet be activated in CNRM-AROME. The developments performed and evaluated in this work demonstrate the 800 benefit in bridging the gap between currently used land surface configurations in AROME (for climate and NWP applications) and existing, state-of-the-art configurations in SURFEX, that require further work in order to be applicable in coupled model experiments and applications.

*Code and data availability.* Météo-France belongs to the ACCORD consortium (http://www.accord-nwp.org/) for development of limited-area models (LAM) and forecasting systems for numerical weather prediction (NWP), within which it cooperates on the development of 805 a shared system of model codes. ACCORD was established in 2021 and initially brought together members of the consortia ALADIN, LACE and HIRLAM. The AROME model forms part of the shared system of model codes. According to the ACCORD Memorandum of Understanding and in particular its Annexes IX and X, all members are allowed to license the shared codes to non-anonymous requests within their home country for non-commercial research. Access to the full AROME code can be obtained by contacting one of the member institutes of the ACCORD consortium.

All computations were performed with Python software version 3.9.13. The codes and snow depth simulations for each CNRM-AROME experiments are available from a zenodo repository (Monteiro et al., 2024). It includes the snow depth simulations for each CNRM-AROME experiments used in the study as well as scripts (in a notebook form) for the following tasks : performing all data preprocessing, reading the different data sources, statistical analyses and figures making.



The remote sensing MODIS (MOD10A1F) dataset is available following this doi : https://doi.org/10.5067/MODIS/MOD10A1F.061.

Part of the in situ snow depth observations were taken from Matiu et al. (2021a) and can be accessed for scientific uses at Matiu et al. (2021b). Additional Austrian snow depth observations are gathered into the TAWES and SNOWPAT datasets and were collected and treated by GeoSphereAT and the Hydrographic Central Office of Austria (HZB). They are accessible for scientific uses upon request to GeoSphereAT. Additional Swiss snow depth observations come from the IMIS datasets (Measurement and IMIS, 2023), accessible for scientific uses.

    The main data from the Col du Lac Blanc data are available at $https : //doi.osug.fr/public/CRYOBSCLIM_CLB/$ and technical in-
formation can be find in Gouttevin et al. (2023). The Torgnon data (Cremonese et al., 2023), metadata and licence information can be accessed here : $https : //meta.icos-cp.eu/objects/40ux_CiuCRP59zo67MrpmM5A$. The Davos dataset from MeteoSwiss can be accessed for scientific uses through the IDAweb portal : https://www.meteosuisse.admin.ch/services-et-publications/service/produits-meteorologiques-et-climatiques/portail-de-donnees-pour-l-enseignement-et-la-recherche.html. The Weissfluhjoch dataset can be accessed for scientific uses from the WSL Institute for Snow and Avalanche Research (SLF).



**Appendix A: Impact of the initialization (spin-up) approach on snowpack simulations**

**(a)** Default intialization fields (01/01/2018)

Snow depth (*m*)

Total water content
of soil ($kg\,m^{-2}$)

Soil temperature at 2m
below surface (°C)

**(b)** Differences between spin-up and default intialization fields (01/01/2018)

Δ Snow depth (*m*)

Δ Total water content
of soil ($kg\,m^{-2}$)

Δ Soil temperature at 2m
below surface (°C)

**Figure A1.** (a) Maps of the snow depth, the total water content of soil and the soil temperature at 2 m fields of our experiments at the initialisation date (01/01/2018) using the default initialisation procedures. (b) Maps of the differences between the default initialisation procedures and the initialisation resulting from 13 years of offline spin-up for the snow depth, the total water content of soil and the soil temperature at 2 m fields at the initialisation date (01/01/2018).





## Simulation of snow depth (*m*) at five sites using default and spin-up initialized simulations

**Figure A2.** Time series of the snow depth at four sites over the 01/01/2018 to 31/12/2020 using the ES-DIF-OPT configuration, either with the default initialisation field (darkgreen continuous line) or the initialisation field resulting from 13 years of offline spin-up (lightred continous line).



**Figure A3.** Time series of the total soil water content at four sites over the 01/01/2018 to 31/12/2020 using the ES-DIF-OPT configuration, either with the default initialisation field (darkgreen continuous line) or the initialisation field resulting from 13 years of offline spin-up (lightred continous line).







**Figure A4.** Time series of the soil temperature at 2 m below surface at four sites over the 01/01/2018 to 31/12/2020 using the ES-DIF-OPT configuration, either with the default initialisation field (darkgreen continuous line) or the initialisation field resulting from 13 years of offline spin-up (lightred continuous line).



## Appendix B: All tested experiments

### B1 Large scale evaluation of snow depth over the 2018-2019 winter

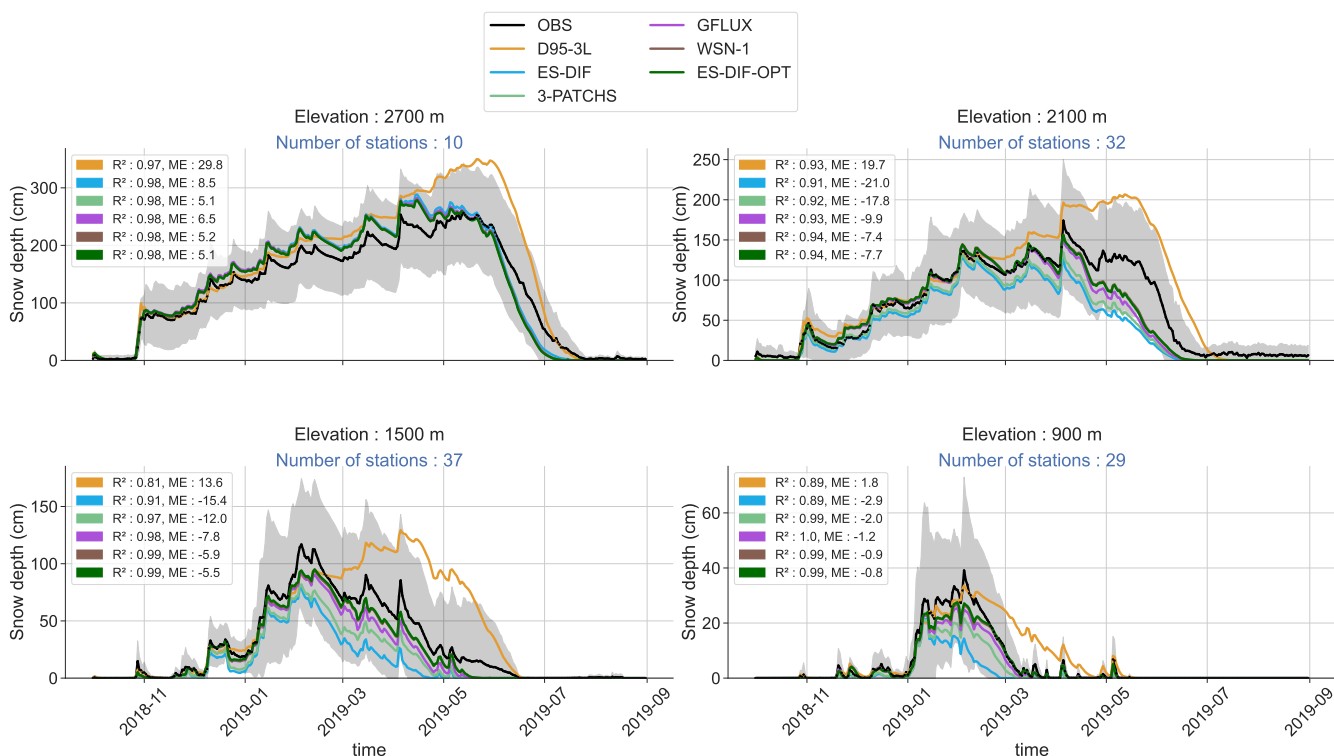

**Figure B1.** Multi-stations mean time series of the height of snow for the 2018-2019 winter for four elevation bands of 300 m (900 m±150 m, 1500 m±150 m, 2100 m±150 m and 2700 m±150 m). Colored continuous lines corresponds to the simulated multi-stations mean time series for each of the configuration. Black lines with circle markers correspond to the multi-stations mean time series of the in situ measurements with the inter-stations standard deviation represented in gray shaded areas. For each elevation bands, the number of stations used to construct the mean and the standard deviation are displayed in blue font. At each elevation bands and for all configurations, the correlation ($R^2$) and the mean error ($ME$) computed using the multi-mean time series between the simulated and the in-situ measurements are displayed.





## B2 Snow cover duration evaluation using MODIS remote sensing data



**Figure B2.** Snow cover duration differences between the different configurations and MODIS observations in the European Alps. Note that MODIS products initially at 500 m horizontal resolution have been reggrided over CNRM-AROME horizontal resolution (2.5 km) grid using a first-order conservative method. **(a)** Map of the average differences (mean error) of the snow cover duration (SCD) over 2 seasons (2018-2019 and 2019-2020) for each configurations compared to MODIS SCD. **(b)** Boxplot representing the spatial distribution of the average differences (mean error) of the SCD over 2 seasons (2018-2019 and 2019-2020) compared to MODIS SCD for each datasets for six elevation bands of 300 m (900 m±150 m, 1500 m±150 m, 1800 m±150 m, 2100 m±150 m, 2400 m±150 m and 2700 m±150 m). Each column corresponds to the values classified by prevailing type of vegetation (see section 3.2 for details).




## Appendix C: Multivariate comparison at four well-instrumented sites

**C1 Location and characteristics of the station and the corresponding CNRM-AROME grid point**

## Location of well-instrumented stations

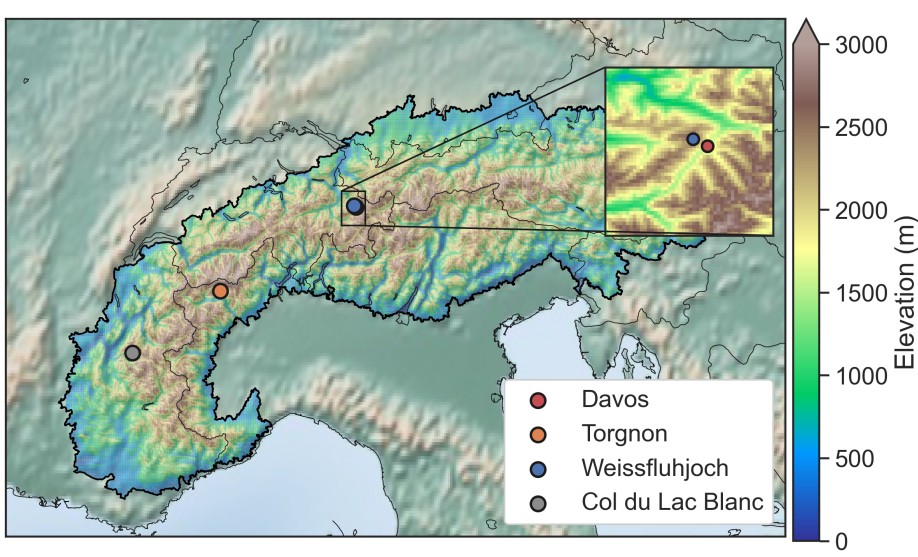

**Figure C1.** Location of the well instrumented in-situ stations together with the digital elevation model at 1 km horizontal resolution

| | Stations | | | CNRM-AROME grid cell | |
|---|---|---|---|---|---|
| **Sites** | **Longitude, latitude (°)** | **Elevation (m)** | **Surface type** | **Elevation (m)** | **Surface type** |
| **Davos (DAV)** | 9.84355, 46.81297 | 1590 | Grassland | 1741 | 66 % high vegetation 33% low vegetation |
| **Torgnon (TOR)** | 7.57805, 45.84444 | 2168 | Grassland | 2299 | 10 % rock 90 % low vegetation |
| **Weissfluhjoch (WFJ)** | 9.80928, 46.82964 | 2536 | Rock | 2407 | 66 % rock 33 % low vegetation |
| **Col du Lac Blanc (CLB)** | 6.11197, 45.12758 | 2720 | Rock | 2738 | 100 % rock |

**Table C1.** Main characteristics of the well-instrumented in-situ stations. Besides their longitude and latitude, the elevation and surface type are given for the stations itself and for the corresponding CNRM-AROME grid cell including it and used for the comparison.



## C2 Comparison of the height of snow, the snow cover duration, and mean errors of multiple variables during snow cover days

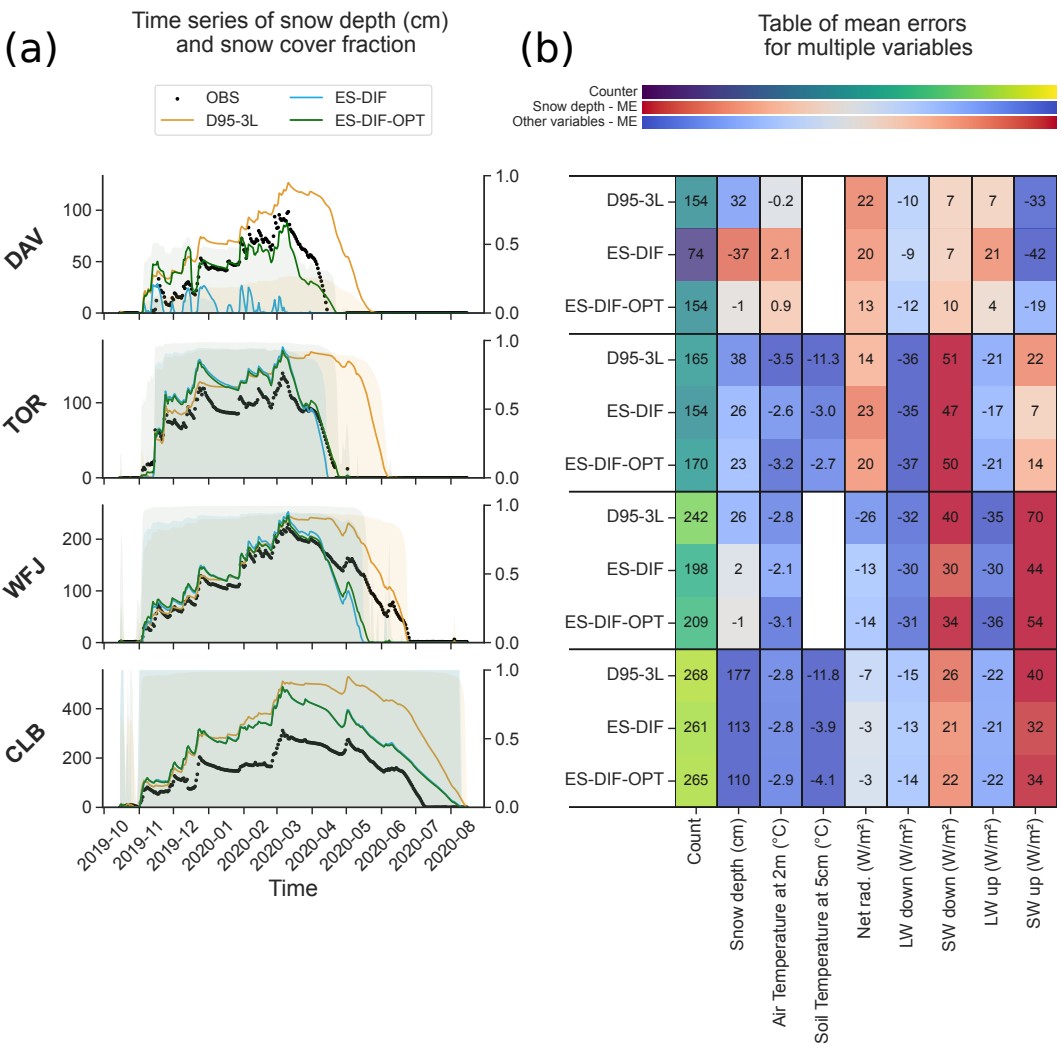

**Figure C2.** a - Panel of time series of 3 variables: snow depth, air temperature at 2 m and net radiation at the four well-instrumented stations (i.e. Davos (DAV), Torgnon (TOR), Weissfluhjoch (WJF) and Col du Lac Blanc (CLB)) for the period 01/10/2019 to 30/09/2020. The colored solid lines correspond to the time series simulated for each land surface configuration at the grid cell including the site location, while the black dots indicate the observed time series. For each graph and each experiment, the simulated snow fraction values are indicated in the colored shaded areas (the y-axis on the right refers to the simulated snow fraction). b - Mean error ($ME$) values for each of the configurations for the four well-instrumented stations. The $ME$ values are calculated over daily values for the 2019-2020 season for days for which snow is present (i.e. snow depth > 1 cm) in both the observed and simulated time series. The "Count" refers to the number of days used to compute the score.



*Author contributions.* DM, CC and SM conceptualized the study. DM performed the visualisation and the formal analyses. DM, SM, AN and ML did the investigation. DM, CC, AN and AA did the data curation. DM and SM designed the methodology. AA, AN and MF provided support with the software. DM, SM, ML, CC and MF wrote the original draft. SM supervised the work.

*Competing interests.*

The contact author has declared that neither of the authors has any competing interests.

*Acknowledgements.* The authors gratefully acknowledge the WCRP-CORDEX-FPS on Convective phenomena at high resolution over Europe and the Mediterranean [FPSCONV-ALP-3]. This work is part of the Med-CORDEX initiative (http://www.medcordex.eu). CNRM/CEN is a member of LabEX OSUG@2020.

We thank the following institutions for sharing with us daily snow depth and surface variables records from station data : the GeopshereAT and specifically Roland Koch and Marc Olefs and sharing with us quality-checked data, the Hydrographic Central Office of Austria (HZB), MeteoSwiss and the WSL Institute for Snow and Avalanche Research (SLF) and specifically Christoph Marty for the contact and sharing with us quality-checked data.

We thank Isabelle Gouttevin and Hugo Mersizen for their valuable advice on the use of Col du Lac Blanc data, Yves Lejeune for his advice on the use of Col de Porte data.

We thank Patrick Samuelsson, Danijel Belusic and Andreas Dobler for their valuable exchanges on HARMONIE-AROME developments and issues.

We thank Ingrid Etchevers for her valuable contributions during the numerous discussions on issues related to the NWP AROME model.



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
