# Peer review of "Improvements of the land surface configuration to better simulate seasonal snow cover in the European Alps with the CNRM-AROME (cycle 46) convection-permitting regional climate model"

_EGUsphere, 2024_

## Referee Comment (RC1)

In addition to being able to develop and advect convective precipitation with the model dynamics, better resolution of topography and snow cover have been proposed as benefits of modelling at "convection permitting" scales. Monteiro et al. investigate the benefits of improvements in the land surface and snow components of a convection-permitting regional climate model. The D95 configuration used as the baseline is far from the state of the art in climate and NWP models, but it is relevant as the land surface model in the current CNRM-AROME regional climate model.

**Abstract**
From the abstract alone, it is not clear what "multiple patches for land surface grid points" means.

**Introduction**
"the use of high-resolution models would minimize modeling uncertainties, by limiting the use of sub-grid parameterizations"; with the exception of convection, none of the subgrid phenomena listed earlier in this paragraph are parametrized in AROME.
"Assessing the representation of snow in coupled configurations is a necessary complementary approach to standalone model run". I would rather put it the other way round: coupled configurations are essential for NWP and climate applications, and it is standalone model runs that can be a useful complement to the evaluation. Raleigh et al. (2015) and Lapo et al. (2015) cited here do not consider coupled surface-atmosphere simulations.

**2.2.1**
A display equation should be integrated into the text as part of a sentence; equations (1) – (6) are not.
"latent L heat flux" should be LE to match equation (4.2).
LWu in equation (3) really is incorrect; it should include reflected LWd if the surface emissivity is not 1 (Kirchoff's law of thermal radiation).
"the atmospheric fluxes received … for all tiles and patches" means specifically the incoming radiation and precipitation fluxes.

**2.3**
The French "expérience" should be translated as "experiment"
"The density, an exponentially decreasing function, forced to 100 kg m$^{-3}$ for fresh snow, limited to 300 kg m$^{-3}$ for aged snow"; density is an increasing function. 300 kg m$^{-3}$ can be low for aged snow and could contribute to overestimates of depth.

**2.4**
The horizontal resolution of ERA5 is 0.25°, not 50 km; was it regridded?

**Figure 3**
The distribution of elevation in the Alpine domain could be added to the bar chart as an indicator of representativeness.

**2.6.2**
State that the MODIS product used is MOD10A1F. How will gap filling and dense vegetation influence uncertainties in snow cover duration?
To determine snow cover duration, a threshold is put on observed snow cover fraction and modelled snow depth, but the model already calculates a snow cover fraction (equations 4 – 6); why not use that? Are the observation and model thresholds consistent?

**2.7.4**
"$\sigma_x$ et $\sigma_y$" are not in the equations. $x$ and $y$ are a model variable and an observation in some order. $r_{xy}$ here becomes $R$ in 3.1.

**3.1**

Figure 6 suggest that ME and $R^2$ are poor indicators of model performance. A low value of ME can be obtained by averaging underestimates and overestimates at different times. The D95-3L simulations are very poor for the melt season, but are given $R^2$ values exceeding 0.8 in all cases.

**3.2**

Observed SCD per elevation band would be a useful complement to Figure 7. Errors can be larger at intermediate elevation simply because there is more room for error when SCD is not close to 0 or 365 days.

**Discussion**

The amount of the Discussion dedicated to discussing results in appendices that have not yet been presented to the reader is odd. If the results are important enough to discuss in detail, they should be presented in sequence in the main text.

**4.2**

ES-DIF $R^2$ is only degraded compared to D95-3L at 2100 m in Figure 6.

Evaluating the ES-DIF configuration that is not actually used in offline or coupled simulations is of limited value; it has already been effectively rejected.

---

## Author Comment (AC1)

**Responses to RC on manuscript EGUSPHERE-2024-249**

***Authors :***
Diego Monteiro, Cécile
Caillaud, Matthieu Lafaysse,
Adrien Napoly, Mathieu
Fructus, Antoinette Alias,
Samuel Morin

**1 Reviewer comments 1 : Richard Essery**

**In addition to being able to develop and advect convective precipitation with the model dynamics, better resolution of topography and snow cover have been proposed as benefits of modelling at convection permitting scales. Monteiro et al. investigate the benefits of improvements in the land surface and snow components of a convection-permitting regional climate model. The D95 configuration used as the baseline is far from the state of the art in climate and NWP models, but it is relevant as the land surface model in the current CNRM-AROME regional climate model.**

\*\*

We thank Richard Essery for the insightful comments and suggestions, which have lead to improving the quality of our manuscript. Below we provide point-by-point answers to the suggestions and comments.

**1.1 Abstract**

**1.1.1 From the abstract alone, it is not clear what multiple patches for land surface grid points means.**

We thank the reviewer for this comment. Indeed, it may not be explicit what multiple patches refers to for readers beyond the SURFEX-ISBA community. Accordingly, we modified the sentence l.11-13 : *"More specifically, the study tests the influence of various changes in the land surface configuration, such as the use of a multilayer soil and snow scheme, the division of the energy balance calculation by surface type within a grid cell (multiple patches), new physiographic databases and parameter adjustments."*

**1.2 Introduction**

We thank Richard Essery for the two following comments concerning the developments of the introduction, for which some formulations can be misleading in their current form.

**1.2.1 the use of high-resolution models would minimize modeling uncertainties, by limiting the use of sub-grid parameterizations ; with the exception of convection, none of the subgrid phenomena listed earlier in this paragraph are parametrized in AROME.**

This sentence did not refer directly to the atmospheric processes listed above, but was intended to emphasize that higher resolution is also of interest for the representation of surface heterogeneities, limiting the need for parameterizations to take into account sub-grid heterogeneities in LSM (Land Surface Model). Nevertheless, the succession of paragraphs can lead to confusion, and consequently sentence L.38-39 has been deleted, and a sentence L.47 has been added : *"In addition to their capacity to explicitly resolve deep convection and thereby enhance the representation of precipitation extremes (Caillaud et al., 2021), models operating at the km-scale make it possible to better represent the topography of mountain areas, and the heterogeneities that characterize the surface through higher resolution, holding great potential for mountain regions."*

**1.2.2 Assessing the representation of snow in coupled configurations is a necessary complementary approach to standalone model run. I would rather put it the other way round : coupled configurations are essential for NWP and climate applications, and it is standalone model runs that can be a useful complement to the evaluation. Raleigh et al. (2015) and Lapo et al. (2015) cited here do not consider coupled surface-atmosphere simulations.**

We agree that the proper way to express the idea would be the other way around. As the sentence can be misleading we decided to rephrased it in the revised manuscript L.31-32 : "Testing snow models in standalone "offline" configurations is not sufficient, and tests in coupled configurations are required. This is particularly challenging in mountainous areas."

With regard to the use of the citations Raleigh et al. (2015) and Lapo et al. (2015) the objective was to provide evidence that, under coupled conditions, atmospheric errors (an unavoidable consequence of using a model) can significantly affect the accuracy of snowpack simulations, even in the absence of coupling. This is precisely what the two articles demonstrate by measuring the impact of errors in atmospheric forcings on snow simulations.

**1.3   2.2.1**

**1.3.1   A display equation should be integrated into the text as part of a sentence ; equations (1)  (6) are not. latent L heat flux should be LE to match equation (4.2). LWu in equation (3) really is incorrect ; it should include reflected LWd if the surface emissivity is not 1 (Kirchoffs law of thermal radiation). the atmospheric fluxes received for all tiles and patches means specifically the incoming radiation and precipitation fluxes.**

We would like to thank Richard Essery for pointing out the formulation errors in the display equations, which have been incorporated into the sentences in the revised version.

We have also corrected the formulation error in equation (3) to take account of the $LWd$ reflected :

$$Rn = SWd + \varepsilon_s LWd - SWu - LWu \tag{1}$$

with $Rn$ the radiation balance ($\mathrm{W\,m^{-2}}$), $SWd$ the incoming solar radiation, $LWd$ the infrared atmospherical radiation, $SWu$ the reflected shortwave radiation and $LWu$ the emitted longwave radiation.

**1.4   2.3**

**1.4.1   The French expérience should be translated as experiment The density, an exponentially decreasing function, forced to 100 kg m3 for fresh snow, limited to 300 kg m3 for aged snow ; density is an increasing function. 300 kg m3 can be low for aged snow and could contribute to overestimates of depth.**

We thank Richard Essery for pointing out the typo, which we have corrected accordingly.

We would also like to thank him for his insightful comment regarding the role of density formulation in the overestimation of simulated snow height. While we have not tested this element, it is a potential factor that may contribute to the modeled overestimation of snow height in the D95-3L configuration. However, our analysis suggests that the primary factor is likely the underestimation of melt, due to his formulation driven by the daily variations of the surface temperature in the presence of vegetation, as discussed in section 4.1.

**1.5   2.4**

**1.5.1   The horizontal resolution of ERA5 is 0.25°, not 50 km ; was it regridded ?**

The ERA5 data have indeed been regridded, and the method employed is quadratic interpolation. The reason for the use of these forcings instead of the native ones is purely technical : they are the highest resolution forcings initially available on the servers at the time the simulations were initiated. Given the time constraints associated with downloading data at native resolution, it was preferable to use those available.

However, the impact of using these instead of finer resolution may be significant, particularly on the formation of fine-scale structures close to the domain boundaries. Indeed, the literature on the subject recommends that the resolution jump for dynamic downscaling should not exceed a factor of 12 (Leduc and Laprise, 2009 ; Leduc et al., 2011). In practice, the greater the jump, the greater the distance from the edges of the domain required for the correct formation of fine-scale structures (i.e., spatial spin-up) to be respected, and therefore the greater the number of points at the edges that will be unusable (Matte et al., 2017).

In the case of our simulation on the ALP-3 domain, the European Alps are located at the center of the domain, several hundred kilometers (more than a hundred grid points) from the domain boundaries, we therefore consider that spatial spin-up is effective and does not affect the results of our simulations.

**1.6   Figure 3**

**The distribution of elevation in the Alpine domain could be added to the bar chart as an indicator of representativeness.**

Following the recommendation of Richard Essery, the frequency distribution of elevation, as represented by a digital elevation model (DEM) at a resolution of 100 m, has been incorporated into the bar chart in

Figure 3 of the revised manuscript. Although we believe that these data can be useful to readers, as they allow for the comparison of the elevational distribution of available observations to the areal elevational distribution of our region of interest, they do not provide direct insights into the representativeness of observation stations. It is important to note that while the elevation distribution of observations may be similar to the frequency distribution of the actual elevations, this does not necessarily indicate an accurate representation of their spatial distributions. This could potentially lead to an over- or under-sampling of certain areas, but it does not provide insight into the reliability of statistics or scores calculated over a range of elevations.

**1.7   2.6.2**

**1.7.1   State that the MODIS product used is MOD10A1F. How will gap filling and dense vegetation influence uncertainties in snow cover duration ? To determine snow cover duration, a threshold is put on observed snow cover fraction and modelled snow depth, but the model already calculates a snow cover fraction (equations 4 6) ; why not use that ? Are the observation and model thresholds consistent ?**

According to (Hall et al., 2019), the gapfilling algorithm replaces the value of pixels categorized as cloud for a given date with the NDSI value of the same pixel for the past nearest date for which it is available. According to the article, values are found in most cases within the previous 3 days (on a study area in the US Rockies), up to 10 days at most. Uncertainties therefore need to be considered on a daily scale regarding NDSI variation for a given pixel (in the case of new snowfall leading to an increase in NDSI, or melting of the snowpack leading to a decrease), which may therefore not be seen.

Working on an aggregated indicator such as snow duration, based on a binary snow/no snow classification determined using a relatively low NDSI value (i.e. 0.2), we expect the impact of daily NDSI uncertainties on snow duration estimation to be small in most cases.

Indeed, if we consider the uncertainties at the beginning and/or end of the season regarding the duration of snow cover, we can expect them to be of the order of 3 to 10 days at most (i.e. order of magnitude of the uncertainties on the daily NDSI). They can therefore be high in the worst cases (from 10 to 30%) for elevation below 1500 m, where the duration of snow cover is of the order of 25-30 days (cf. figure 1), but do not exceed 10% in the most unfavorable cases at 1500 m and above (median duration of snow cover of the order of 115 days, cf. figure 1).

[Figure]

FIGURE 1 – Boxplot representing the spatial distribution of the snow cover duration values for two seasons (2018-2019 and 2019-2020) for multiple elevation bands in the European Alps.

Regarding the uncertainties in forest areas, it's true that, since they are based on optical detection, the masking effect of the canopy attenuates the signal and thus the NDSI as compared to an equivalent quantity of snow in an open area. Several studies have evaluated MODIS products from the NSIDC Collection 5, directly presented as a binary (value (0/1)) indicating for each pixel the presence or absence of snow, constructed from an NDSI threshold, variable between 0.1 and 0.4 as a decreasing function of NDVI (spectral index depending on the presence of vegetation). On these products, Parajka et al. (2012) was able to show by comparison with in situ snow depth measurements in a mountainous watershed in Slovakia that they maintain an accuracy score of over 81% for mixed forest and open areas, and close to 92% for forest-only areas, compared with 98% in open areas. Gascoin et al. (2015) in a Pyrenean watershed show accuracies higher than 90% in forested areas when compared with in situ snow height measurements and satellite observations at higher resolutions (i.e. Landsat 5 and 7). We therefore have good confidence in the ability of MODIS sensors to detect snow under forest cover, provided that we lower the NDSI threshold.

In our study, we used Collection 6 of the NSIDC MODIS product, in which only NDSI values are provided. A snow presence/absence binary variable was constructed, using an NDSI threshold of 0.2 for all areas, corresponding to a relatively low snow fraction (of the order of 20-30% according to Salomonson and Appel (2004)) to facilitate the detection of snow beneath the forest. In a previous article (Monteiro and Morin, 2023), we were able to evaluate the product by comparing the absence and presence of snow in MODIS pixels with in situ snow depth measurements, taking different snow depth thresholds for a given NDSI threshold. Several scores on the duration snow cover were calculated, as well as the confusion matrix based on the absence and presence of snow. The accuracy scores showed values above 90% for snow depth thresholds ranging from 1 to 5 cm, and the differences in snow cover duration show that the 1 cm threshold minimizes mean absolute errors.

It should also be noted that we use these snow cover durations as comparative values (order of magnitude) and not as absolute references, which we should approach as closely as possible. As the differences between the simulated snow cover durations and those calculated from satellite observations are very large compared with the uncertainties of the snow cover durations estimated by satellite observations, we consider that these errors have little impact on the main conclusions of our study.

Regarding the last part of the question, we have decided not to use the snow fraction as calculated by

the model, as we consider that the function which determines it is much more an adjustment variable used to modulate fluxes at the soil-atmosphere interface (which should be modified in the future), than a realistic estimate of the snow cover fraction within a pixel.

**1.8   2.7.4**

**1.8.1   $\sigma_x$ et $\sigma_y$ are not in the equations. $x$ and $y$ are a model variable and an observation in some order. $r_{xy}$ here becomes $R$ in 3.1.**

We thank Richard Essery for his remarks, the manuscript have been corrected accordingly.

**1.9   3.1**

**1.9.1   Figure 6 suggest that ME and $R^2$ are poor indicators of model performance. A low value of ME can be obtained by averaging underestimates and overestimates at different times. The D95-3L simulations are very poor for the melt season, but are given $R^2$ values exceeding 0.8 in all cases.**

We agree with Richard Essery that these scores exhibit a number of limitations.

In the case of the mean error, it is possible to obtain a score close to zero, despite the presence of over- and underestimation of a similar magnitude. This phenomenon occurs when the two types of estimation offset each other, resulting in a net zero error. Nevertheless, the mean error remains an relevant score that can highlight systematic biases (over- and/or under-estimation) by providing a straightforward and easily comprehensible indication of the sign of differences.

In the specific case of this study, where the differences are systematic in the vast majority of cases considered, the score can be used to quantify (in conjunction with the simulated and observed snow depth curves) the amplitude of overestimations. Concerning the $R^2$, in our opinion it illustrates well, in the same way as the mean error, the differences we wish to highlight between the curves. As Richard Essery mentionned, the $R^2$ remains high (always above 0.8) even in the case of the D95-3L configuration, where a clear discrepancy at the end of the season can be observed. Nevertheless, its high score serves as a reminder that, until this late-season discrepancy, the snow model's behavior regarding the accumulation and evolution during the winter months remains satisfactory, and for all the configurations tested.

We have therefore chosen to retain these two indicators in the revised version of the manuscript.

**1.10   3.2**

**1.10.1   Observed SCD per elevation band would be a useful complement to Figure 7. Errors can be larger at intermediate elevation simply because there is more room for error when SCD is not close to 0 or 365 days.**

We thank Richard Essery for his relevant remark pointing out a potential factor to explain the largest differences of the snow cover duration that we obtain at intermediate elevations.

As shown in Figure 1, over the four elevation bands investigated, the variance in snow cover duration is highest at intermediate elevations (i.e. 1500 m). However, it is significantly reduced at 2100 m and 2700 m for both seasons. Nevertheless, the errors calculated for the different configurations are in many cases similar or even greater for the 2100 m elevation band than for the 1500 m elevation band.

Although we believe this may be a factor that mechanically widens errors at intermediate altitudes, it probably acts to second order compared to the presence of a partial snow cover fraction exacerbated by the presence of vegetation as discussed in section 4.2 of the manuscript.

Based on these remarks, we have decided to add the distribution of snow cover duration values for each of the two years in Figure 4 of the revised manuscript.

**1.11   Discussion**

**1.11.1   The amount of the Discussion dedicated to discussing results in appendices that have not yet been presented to the reader is odd. If the results are important enough to discuss in detail, they should be presented in sequence in the main text.**

We would like to thank Richard Essery for his valuable insights on the content of the appendices and the discussion. Consequently, figure C2 in appendix C has been relocated to the discussion section, accompanied by a more comprehensive description of the figure in the revised manuscript. We believe these amendments will have a beneficial impact on section 4.4, enhancing the clarity and accessibility for readers.

**1.12   4.2**

**1.12.1   ES-DIF $R^2$ is only degraded compared to D95-3L at 2100 m in Figure 6. Evaluating the ES-DIF configuration that is not actually used in offline or coupled simulations is of limited value; it has already been effectively rejected.**

We would like to thank Richard Essery for his comments on the discussion of the ES-DIF configuration results.

With regard to the first remark concerning the degradation of the $R^2$ score between the ES-DIF and D95-3L configurations, which is only present at 2100 m and not at all elevations, this was a typo error that has been corrected in the revised manuscript.

However, we do not share the referee's view expressed in the second comment that the ES-DIF configuration is not used in offline and coupled configurations and that its evaluation would therefore be of limited value. It is true that, as presented, the ES-DIF configuration is not currently used in NWP nor climate contexts. However, the ES multi-layer snow model and the DIF soil model are used in many contexts, from global climate simulation with CNRM-CM6 (Decharme et al., 2019) to high-resolution regional simulations with the HARMONIE-AROME model (Belušić et al., 2020). It's also true that this configuration has already been evaluated with weaknesses pointed out by (Napoly et al., 2020; Nousu et al., 2023), but in a stand-alone point-scale evaluation. To the best of our knowledge, our study is the only one to evaluate such configuration in a coupled, spatialized at high-resolution context in a mountain region. In addition, autonomous evaluation of the ES-DIF configuration (i.e. without the adjustments made in ES-DIF-OPT) makes it possible to document its weaknesses and iteratively test the various modifications needed to mitigate them.

**Références**

Belušić D, de Vries H, Dobler A, Landgren O, Lind P, Lindstedt D, Pedersen RA, Sánchez-Perrino JC, Toivonen E, van Ulft B, Wang F, Andrae U, Batrak Y, Kjellström E, Lenderink G, Nikulin G, Pietikäinen JP, Rodríguez-Camino E, Samuelsson P, van Meijgaard E, Wu M (2020) Hclim38 : a flexible regional climate model applicable for different climate zones from coarse to convection-permitting scales. Geoscientific Model Development 13(3) :1311–1333, DOI 10.5194/gmd-13-1311-2020, URL https://gmd.copernicus.org/articles/13/1311/2020/

Caillaud C, Somot S, Alias A, Bernard-Bouissières I, Fumière Q, Laurantin O, Seity Y, Ducrocq V (2021) Modelling mediterranean heavy precipitation events at climate scale : an object-oriented evaluation of the cnrm-arome convection-permitting regional climate model. Climate Dynamics 56(5) :1717–1752, DOI 10.1007/s00382-020-05558-y

Decharme B, Delire C, Minvielle M, Colin J, Vergnes JP, Alias A, Saint-Martin D, Séférian R, Sénési S, Voldoire A (2019) Recent changes in the ISBA-CTRIP land surface system for use in the CNRM-CM6 climate model and in global off-line hydrological applications. Journal of Advances in Modeling Earth Systems 11(5) :1207–1252

Gascoin S, Hagolle O, Huc M, Jarlan L, Dejoux JF, Szczypta C, Marti R, Sánchez R (2015) A snow cover climatology for the pyrenees from modis snow products. Hydrology and Earth System Sciences 19(5) :2337–2351, DOI 10.5194/hess-19-2337-2015

Hall DK, Riggs GA, DiGirolamo NE, Román MO (2019) Evaluation of modis and viirs cloud-gap-filled

snow-cover products for production of an earth science data record. Hydrology and Earth System Sciences 23(12) :5227–5241

Lapo KE, Hinkelman LM, Raleigh MS, Lundquist JD (2015) Impact of errors in the downwelling irradiances on simulations of snow water equivalent, snow surface temperature, and the snow energy balance. Water resources research 51(3) :1649–1670, DOI 10.1002/2014WR016259

Leduc M, Laprise R (2009) Regional climate model sensitivity to domain size. Climate Dynamics 32 :833–854

Leduc M, Laprise R, Moretti-Poisson M, Morin JP (2011) Sensitivity to domain size of mid-latitude summer simulations with a regional climate model. Climate dynamics 37 :343–356

Matte D, Laprise R, Thériault JM, Lucas-Picher P (2017) Spatial spin-up of fine scales in a regional climate model simulation driven by low-resolution boundary conditions. Climate Dynamics 49 :563–574

Monteiro D, Morin S (2023) Multi-decadal analysis of past winter temperature, precipitation and snow cover data in the european alps from reanalyses, climate models and observational datasets. The Cryosphere 17(8) :3617–3660, DOI 10.5194/tc-17-3617-2023, URL `https://tc.copernicus.org/articles/17/3617/2023/`

Napoly A, Boone A, Welfringer T (2020) Isba-meb (surfex v8. 1) : model snow evaluation for local-scale forest sites. Geoscientific Model Development 13(12) :6523–6545

Nousu JP, Lafaysse M, Mazzotti G, Ala-aho P, Marttila H, Cluzet B, Aurela M, Lohila A, Kolari P, Boone A, et al. (2023) Modelling snowpack dynamics and surface energy budget in boreal and subarctic peatlands and forests. EGUsphere 2023 :1–52

Parajka J, Holko L, Kostka Z, Blöschl G (2012) Modis snow cover mapping accuracy in a small mountain catchment–comparison between open and forest sites. Hydrology and Earth System Sciences 16(7) :2365–2377

Raleigh M, Lundquist J, Clark M (2015) Exploring the impact of forcing error characteristics on physically based snow simulations within a global sensitivity analysis framework. Hydrology and Earth System Sciences 19(7) :3153–3179

Salomonson VV, Appel I (2004) Estimating fractional snow cover from modis using the normalized difference snow index. Remote sensing of environment 89(3) :351–360, DOI 10.1016/j.rse.2003.10.016

---

## Author Comment (AC2)

**Responses to RC on manuscript EGUSPHERE-2024-249**

***Authors :***
Diego Monteiro, Cécile
Caillaud, Matthieu Lafaysse,
Adrien Napoly, Mathieu
Fructus, Antoinette Alias,
Samuel Morin

**1   Reviewer comments : Emanuel Dutra**

**The study presents an improved land surface snow representation in the CNRM-AROME model evaluated over the European Alps at convection-permitting resolution (2.5km). The snow simulations are evaluated against a large sample of in-situ snow depth observations and remote sensing snow cover. The manuscript is well written with good quality graphics and with a very detailed and interesting process-oriented discussion of the results. In my opinion, this manuscript is of interest to the community and fits well with GMD scope. However, there are several decisions on the manuscript organization and results presentation that, in my opinion, limit the main message of the study, and my suggestion would be for the authors to consider some re-organization of the results presentation and discussion, along with a few minor clarifications listed bellow.**

**

We would like to thank Emanuel Dutra for his constructive evaluation of the article. His assessment is positive and detailed, and we appreciate his time in evaluating our work and providing a set of comments for improvement. We have provided a point-by-point response to his comments below.

**1.1   ES-DIF should not be an experiment but a sensitivity test, It is mentioned in section 2.3.2 : However, note that only one patch is used herein for the NATURE tile, which is not the way the configuration is implemented for the coupled systems CNRM-CM6 and CNRM-ALADIN using 12 patches, and HARMONIE-Climate using 2 patches. and also in the discussion We also note that there are little to no cases where the ES-DIF configuration, with only one patch, has been implemented in an offline or coupled modeling system. I would suggest to re-organize partially the results/discussion having two configurations : D95-3L and ES-DIF-OPT, where ES-DIF-OPT is presented and D95-3L used as benchmark, while ES-DIF, 3-PATCHS, GFLUX, WSN-1 are sensitivity experiment to allow a process-oriented discussion. For example, in my opinion, Figures B1 and B2 are the most interesting results of this study, but are presented in the appendix.**

We thank Emanuel Dutra for his suggestions. His remarks tie in with the more general consideration of editorial choices made in the course of writing an article, particularly in the case of research results containing a large number of hypotheses and tested experiments, and the way in which these are to be told and developed.

For this article, we have chosen to focus on three main configurations so as not to overload the reader with a complex set of experiments, allowing us to easily convey the main results we wish to highlight. There are three main results we wanted to focus on :

— The initial configuration doesn't allow to reproduce the seasonal evolution of the snow cover in the European Alps, particularly in areas where vegetation is present. This is due to the lack of an explicit representation of the snowpack and the vegetation (both included in a composite surface layer), and to the way in which the surface energy balance is accounted for in these cases to melt the snowpack.

— A configuration that includes only an enriched representation of the soil and an explicit representation of the snowpack also fails to simulate the seasonal evolution of the snowpack satisfactorily, despite variations in snow height that are closer to observational references than the reference configuration. These limitations are directly linked to the way in which energy balances are calculated when there is a partial snow fraction (this happens much more frequently in the presence of high vegetation, due to the formulation of the snow cover fraction under vegetation).

— A final configuration (including a set of modifications) based on the previous configuration enables to significantly reduce deviations from references by significantly improving the seasonal evolution of snowpack. The modifications tested iteratively allow us to identify the main factors affecting the representation of the snowpack in the ES-DIF.

For this purpose, we have chosen to present the standard configuration used here as a reference, since it is the one used in the previous version of the CNRM-AROME model, and to present the final configuration, since it is the most satisfactory in terms of the metrics evaluated for the representation of snow cover in the European Alps.

We also decided to include the ES-DIF configuration, despite the fact that it is little used in the literature. The choice of this intermediate configuration was based on multiple criteria :

— The distance of this configuration from the others in terms of modifications to the individual elements of the configuration, as well as in terms of snow simulation results.

— The fact that it allows us to highlights difficulties in the representation of the snowpack that are specific to the way energy exchanges at the ground-snow interface are handled (cf. section 4.2). Although the shortcomings of this configuration have already been identified in previous standa-lone simulation at the point-scale, this study is the first large-scale, coupled surface-atmosphere, spatialized and at high-resolution evaluation of this configuration, and we believe that it offers real added value in terms of its presentation in the body of the text.

— It also underlines the fact that the sudden addition of enriched physical components does not automatically lead to improved metrics, and highlights the importance of the way in which the various components interface and the representation of sub-grid heterogeneities.

For these reasons, ES-DIF appears to us as an intermediate experiment facilitating the understanding of the main messages of the study, and we have chosen in the revised manuscript to keep this experiment in the main body of the text.

The elements communicated above on the main messages of the study also explain why figures B1 and B2 have been placed in appendices. The current organization, which is the result of a choice made with a view to keeping the main body of the text as concise as possible while preserving the main messages of the article, seems to us to be the most optimal. Readers interested in the detailed results of each experiment will find the information in the appendices, and will be able to understand these figures quickly since they are presented in a very similar form in the main body of the text.

Nevertheless, as the reviewer Richard Essery pointed out in his comments too, the last analysis in the appendices C concerning a multivariate comparison at the station appears for the first time in the discussion with almost no introductory sentence to properly apprehend it. To make it easier to understand, we have added it to the body of the text in section 4.4 of the discussion, and provided a brief description to clarify its use in the rest of the paragraph.

**1.2   The 3-PATCHS : The decision to use 3-patches seems a bit unclear. Why not 2 or 4 or 5 ? Just computational cost ? I understand that this is likely driven also by expert knowledge that is difficult to justify. However, the use of 3-patches, representing bare-soil, low-vegetation, high vegetation has been the approach taken by ECMWF since early 2000s already in ERA-40 (https ://doi.org/10.21957/9aoaspz8) more than 20 year ago.**

We would like to thank Emanuel Dutra for his question and remark concerning the choice of the number of patches. As explained in section 2.3.3, this is a trade-off between the representation of the main sub-grid heterogeneities that can be found in three main categories (no vegetation, low vegetation, high vegetation) and the additional computational, writing and storage cost involved in splitting each surface by three. Using a few additional patches would entail additional costs for few benefits (only allowing to separate between high vegetation types), while using only 2 would no longer allow us to distinguish between areas where vegetation is present and those where it is absent. The use of three patches then appeared to us to be the most appropriate and minimal number for the highest improvements.

**1.3   GLUX : The proposed cap of the conductivity at 5% below a snow fraction of 75% and then linear is a pragmatic approach (mentioned in the discussion), which is an acceptable justification, but more importantly it supports the hypothesis of an ove-restimation of the ground heat flux, inducing basal melt in partially snow-covered surfaces. The harmonic average between snow/soil conductivities does not account for the air trapped between the snow base and soil top due to living/dead organic matter that will effectively reduce heat conductivity. This is difficult to represent due to the very high spatial variability of such conditions, but I suggest that this is also discussed as a possible explanation for the overestimation of the ground heat flux.**

We thank Emanuel Dutra for his suggestion on the potential role of the air trapped by vegetation at the soil-snow interface that can act as a heat conductivity reducer. In all the configurations tested here, this effect is not implemented, and it could be a factor contributing to the overestimation of the heat flux. It could act jointly with other factors not taken into account or poorly modeled, which could lead to a reduction in soil thermal conductivity, such as an underestimation of the organic matter content in the soil

surface layers. This is the reason behind the modification of the input databases for soil organic matter content and the activation of its effect, although this had only a negligible effect on the representation of snow cover in our region of interest.

We also note that the effect on thermal conductivity of a layer of organic matter at the soil-snow interface is already available within the SURFEX MEB (Multi Energy Balance) explicit vegetation module (Napoly et al., 2017 ; Boone et al., 2017) and represents a future prospect for coupling with the CNRM-AROME model.

According to this suggestion, we have introduced a sentence at the end of section 4.2 in the revised version of the manuscript to mention the potential effect of factors that would lead the configuration to overestimate thermal conductivity at the soil-snow interface : *"It is important to note that, in addition to the heating feedback from snow-free surfaces to snow-covered surfaces, other factors not directly tested in this study are likely to exacerbate this effect. Indeed, due to their effects on reducing thermal conductivity, some physical processes, such as air trapping at the soil-snow interface due to the presence of a litter layer and/or low vegetation, as well as a poor representation of organic matter content in the upper soil layers, may contribute to and exacerbate basal snowpack melting."*

**1.4   Impact of initialization : I do not see any reason for the authors not to use the spin-up simulation ? Although it has a reduced impact for the actual period of validation, if the authors have a simulation with land initial conditions that are much better, why not use it ?**

We would like to thank Emanuel Dutra for pointing out that we do not use an initial surface state resulting from long spin-up initialization for our experiments. Actually, we would also have preferred to run the simulations with a correctly initialized surface, but technical and computational cost considerations constrained our choice.

In fact, the spin-up was initially carried out over few months for the D95-3L configuration, with little impact on the initial soil conditions, since a "force-restore" scheme was in use, thus requiring no particular spin-up. The implementation of the new ES-DIF configuration using a multi-layer soil with explicit diffusion implies a spin-up of at least 5 to 10 years to reach a point of equilibrium in terms of integrated water and heat content of the soil columns (Christensen, 1999 ; Cosgrove et al., 2003). The main problem is that the numerical cost of the simulations (very high in the case of simulation with the CNRM-AROME model on the ALP-3 domain) limited the realization of a coupled surface-atmosphere spin-up over several years.

In order to avoid the high numerical cost of a coupled spin-up, we decided to perform a standalone simulation of the surface, forced by atmospheric fields from previous CNRM-AROME simulations carried out as part of CORDEX's Convection FPS (Coppola et al., 2020 ; Pichelli et al., 2021 ; Caillaud et al., 2021). However, these standalone simulations were only set up after most of the experiments presented in the study had been carried out. So as not to have to run all the experiments again, and to be able to maintain equivalent initial conditions for all the experiments, we decided to continue with the default initialization, while evaluating the impact on snow cover of a surface with a long spin-up. The differences both initializations being negligible, as shown in section 2.5, we believe that, although not optimal, the lack of a long spin-up does not affect the results and conclusions of our study. We also considered that the added-value of running the simulations again was not worth the resources required to run the experiments on the High-Performance Computing (HPC) infrastructure.

**1.5   Main result not much explored/discussed : The sensitivity results in Figure B1, in particular the activation of the 3-patches shows that even at convection-permitting resolution of 2.5km representing the sub-grid scale variability of land surface heterogeneities of land cover are fundamental. I think that this is a very interesting result in particular within the current European destination earth program, showing that increasing horizontal resolution is not enough for a better representation of the land surface processes mostly due to the very high spatial scale of surface heterogeneities that impact the mean state over the grid-box. I would suggest the author discuss a bit on this in the conclusions and even mention it in the abstract, if they agree.**

We thank Emanuel Dutra for his remark and suggestion. We agree that this is an interesting results of the study. Even at kilometric resolutions, at least in the case of CNRM-AROME, doing without a representation of sub-grid heterogeneities of the surface implies numerous errors, particularly concerning

the estimation of the grid-average energy balance, with a significant effect on the simulation of snow cover in mountain areas.

It should be noted, however, that while we believe these conclusions to be extendable to other surface variables and other regions, the effect of representing sub-grid heterogeneities is in the specific case of snow simulation exacerbated by poor heat flux management due to the assumption of a shared soil between snow-covered and snow-free areas. In the absence of this assumption, with, for example, two separate soil columns for snow-covered and snow-free surface, we can expect a considerable reduction in the overestimation of basal heat flux and a clear improvement of the snow cover simulation, even using only one patch.

Based on the suggestion of Emanuel Dutra, we highlighted this result in the conclusion L.780-785 in the revised manuscript :   *"Their effectiveness confirms the hypotheses put forward to explain the exaggerated melting in the ES-DIF simulation and underlines the importance, even at kilometer resolution, of taking into account the main sub-grid heterogeneities concerning surface type in mountainous terrain. In the analysis of the preferred configuration ES-DIF-OPT, this allows the simulation of snow cover to be satisfactory compared with the references used in many cases."*

And a sentence was modified in the abstract L.15-17 in the revised manuscript :  *"These limitations are addressed in further configurations that highlight the importance, even at kilometer resolution, of taking into account the main sub-grid surface heterogeneities and improving representations of interactions between fractional snow cover and vegetation."*

**1.6    Line 84 : e.g. HTESSEL (Balsamo et. al. 2009) : Suggest to change to ECLand (https ://www.mdpi.com/2073-4433/12/6/723) as this is an updated reference for the ECMWF model that also has a multilayer snow scheme (https ://doi.org/10.1029/2019MS001725)**

The manuscript has been revised accordingly.

**1.7    Lines 87-88 : The sentence The identify shortcomings may also explained some of the snow cover issues raised in  is very unclear, please rephrase it.**

The sentence have been rephrased in the revised manuscript :  *"Moreover, the factors proposed in the study to explain the erroneous representation of the snowpack in CNRM-AROME are strongly suspected to contribute to the shortcomings in the representation of seasonal snow cover documented in coarser resolution coupled simulations using SURFEX-ISBA LSM, such as CNRM-ALADIN (Termonia et al., 2018) in the Alps (Monteiro and Morin, 2023) and CNRM-CM6 in high-latitude boreal forests (Decharme et al., 2019)."*

**1.8    Eq. (5) : Please provide in text (or table) and typical z0 values in the presence of high vegetation used in the model to help understand actual behavior of the parameterization of snow cover fraction in these situations. THis also links with the WSN factor change from 5 to 1. Using a typical z0 value, showing a figure of the snow cover fraction as a function of snow depth with WSN=5 and WSN=1 would be illustrative.**

We thank Emanuel Dutra for his suggestion on additional figures and tables that will help interested readers to understand the behaviour of the snow cover fraction parameterization over vegetation.

The roughness length involved in the formulation is that of the average roughness length over the nature tile for a given grid cell, meaning that it includes not only the vegetation but results from an aggregation of all surface types inside the grid cell, snow included. Therefore its value varies along the winter season, with the amount of snow on the ground, and monthly with the LAI of the vegetation.

Figure 1, added in appendix A in the revised manuscript is a map of the mean surface roughness length values over the two winter seasons (November to April for the 2018-2019 and 2019-2020 periods) using the D95-3L configuration.

[Figure]

FIGURE 1 – **(a)** Map of the mean surface roughness length values over the two winter seasons (November to April for the 2018-2019 and 2019-2020 periods) using the D95-3L configuration. **(b)** Boxplot representing the spatial distribution of the mean roughness length values over the two winter seasons (November to April for the 2018-2019 and 2019-2020 periods) using the D95-3L configuration classified by prevailing type of surface (see section **??** for details). It is noteworthy that, although the roughness length values are displayed for configuration D95-3L and vary with the amount of snow on the grid cell, they are very similar for all the configurations tested.

Figure 2, added in appendix A in the revised manuscript show the snow cover fraction as a function of snow depth for two combinations of different roughness length $z0$ at a given scaling factor $Wsn$, and two combination of the scaling factor $Wsn$ for a given roughness length $z0$.

[Figure]

FIGURE 2 – Snow cover fraction over vegetation as a function of snow depth for multiple combination of values for the roughness length $z0$ and the scaling factor $Wsn$. The black curves represent the sensitivity of the snow cover fraction parameterization using a value of $Wsn = 5$ as is the case for the D95-3L and ES-DIF configuration, while the orange curves a value of $Wsn = 1$, used for the ES-DIF-OPT configuration.

**1.9   Table 1 : Table presents in the last line computational time relative to D95-3L I must admit that I was very surprised to see such an increase when activating ES-DIF (+15%), and even more with -OPT just with 3 patches. In NWP/climate models the land surface component is normally a rather negligible part in the computational cost due to the simplicity of the calculations and to the 1D nature that typically fits very well MPI /OpenMP implementations. This is mostly a curiosity, but do the authors have some explanation for such a significant computational cost increase ?**

We thank Emanuel Dutra for his question concerning the increased computational cost of the successive changes of the land surface configuration.

Indeed, the increased computational cost of the surface is small compared to the absolute computational cost of the atmospheric part. It turns out that, if we look more closely, the dominant factor accounting for most of the increase lies in the time steps used to write the model outputs (every hour). Although the computational overhead is not high, the use of multi-layer schemes for snow and soil, followed by the activation of three patches, multiplies the number of prognostic and diagnostic variables written each hour. Each hour, when switching to the ES-DIF configuration, the number of prognostic variables for soil alone is multiplied by 7 (from 2 to 14 layers), while that for snow is multiplied by 12 (from 1 to 12 layers). In addition, all prognostic and diagnostic variables are multiplied by three when 3 patches are activated. This highlights that increasing the realism of the land surfaces, hence the number of variables, could be limited for some applications by input/output considerations such as the writing of prognostic and diagnostic files.

However, the values shown in Table 2 of the original manuscript may not reflect the additional computational cost that will be of interest to readers, given that input/output processing can be considerably optimized and is workflow-dependent. In the revised manuscript, the computational costs have been replaced to take into account only the model execution time step (excluding input/output considerations).

**1.10   Line 424 ; sigma standard deviation is not used in the equations, no need to define it**

The manuscript has been revised accordingly.

**Références**

Boone A, Samuelsson P, Gollvik S, Napoly A, Jarlan L, Brun E, Decharme B (2017) The interactions between soil–biosphere–atmosphere land surface model with a multi-energy balance (isba-meb) option in surfexv8–part 1 : Model description. Geoscientific Model Development 10(2) :843–872

Caillaud C, Somot S, Alias A, Bernard-Bouissières I, Fumière Q, Laurantin O, Seity Y, Ducrocq V (2021) Modelling mediterranean heavy precipitation events at climate scale : an object-oriented evaluation of the cnrm-arome convection-permitting regional climate model. Climate Dynamics 56(5) :1717–1752, DOI 10.1007/s00382-020-05558-y

Christensen OB (1999) Relaxation of soil variables in a regional climate model. Tellus A 51(5) :674–685

Coppola E, Sobolowski S, Pichelli E, Raffaele F, Ahrens B, Anders I, Ban N, Bastin S, Belda M, Belusic D, et al. (2020) A first-of-its-kind multi-model convection permitting ensemble for investigating convective phenomena over Europe and the Mediterranean. Climate Dynamics 55(1) :3–34

Cosgrove BA, Lohmann D, Mitchell KE, Houser PR, Wood EF, Schaake JC, Robock A, Sheffield J, Duan Q, Luo L, et al. (2003) Land surface model spin-up behavior in the north american land data assimilation system (nldas). Journal of Geophysical Research : Atmospheres 108(D22)

Decharme B, Delire C, Minvielle M, Colin J, Vergnes JP, Alias A, Saint-Martin D, Séférian R, Sénési S, Voldoire A (2019) Recent changes in the ISBA-CTRIP land surface system for use in the CNRM-CM6 climate model and in global off-line hydrological applications. Journal of Advances in Modeling Earth Systems 11(5) :1207–1252

Monteiro D, Morin S (2023) Multi-decadal analysis of past winter temperature, precipitation and snow cover data in the european alps from reanalyses, climate models and observational datasets. The Cryosphere 17(8) :3617–3660, DOI 10.5194/tc-17-3617-2023, URL https://tc.copernicus.org/articles/17/3617/2023/

Napoly A, Boone A, Samuelsson P, Gollvik S, Martin E, Seferian R, Carrer D, Decharme B, Jarlan L (2017) The interactions between soil–biosphere–atmosphere (isba) land surface model multi-energy balance (meb) option in surfexv8–part 2 : Introduction of a litter formulation and model evaluation for local-scale forest sites. Geoscientific Model Development 10(4) :1621–1644

Pichelli E, Coppola E, Sobolowski S, Ban N, Giorgi F, Stocchi P, Alias A, Belušić D, Berthou S, Caillaud C, et al. (2021) The first multi-model ensemble of regional climate simulations at kilometer-scale resolution part 2 : historical and future simulations of precipitation. Climate Dynamics 56(11) :3581–3602, DOI 10.1007/s00382-021-05657-4

Termonia P, Fischer C, Bazile E, Bouyssel F, Brožková R, Bénard P, Bochenek B, Degrauwe D, Derková M, El Khatib R, Hamdi R, Mašek J, Pottier P, Pristov N, Seity Y, Smolíková P, Španiel O, Tudor M, Wang Y, Wittmann C, Joly A (2018) The ALADIN System and its canonical model configurations AROME CY41T1 and ALARO CY40T1. Geoscientific Model Development 11(1) :257–281, DOI 10.5194/gmd-11-257-2018

---

## Author Response (AR2)

**Technical comments from Richard Essery**

The authors have addressed my review comments. One error that I missed in the first review is that I think that the left-hand sides of equations 8 and 12 should be Ct, not 1/Ct, to be dimensionally correct; compare with equation 16 of D95. Otherwise, the revised manuscript is scientifically acceptable. Some copy editing to correct English errors will be required before publication.

→ The manuscript has been corrected according to the equations found in the original publication of Douville et al., (1995).

**Comments from the topical editor**

For my part, I am asking you to check that all links and DOIs to code & data are still up to date after the revisions.

→ A new version of the zenodo repositories has been created and references have been updated in the manuscript accordingly.